# Viral transduction of primary human lymphoma B cells reveals mechanisms of NOTCH-mediated immune escape

Maurizio Mangolini[1,2], Alba Maiques-Diaz [3], Stella Charalampopoulou[3], Elena Gerhard-Hartmann[4], Johannes Bloehdorn [5], Andrew Moore [1,2], Giorgia Giachetti[1,2], Junyan Lu [6,7], Valar Nila Roamio Franklin[8], Chandra Sekkar Reddy Chilamakuri[8], Ilias Moutsopoulos [1], Andreas Rosenwald[4], Stephan Stilgenbauer[5], Thorsten Zenz [9,10], Irina Mohorianu[1], Clive D'Santos[8], Silvia Deaglio [11], Daniel J. Hodson [1,2], Jose I. Martin-Subero [3,12] & Ingo Ringshausen [1,2] ✉

Hotspot mutations in the PEST-domain of *NOTCH1* and *NOTCH2* are recurrently identified in B cell malignancies. To address how *NOTCH*-mutations contribute to a dismal prognosis, we have generated isogenic primary human tumor cells from patients with Chronic Lymphocytic Leukemia (CLL) and Mantle Cell Lymphoma (MCL), differing only in their expression of the intracellular domain (ICD) of NOTCH1 or NOTCH2. Our data demonstrate that both NOTCH-paralogs facilitate immune-escape of malignant B cells by up-regulating PD-L1, partly dependent on autocrine interferon-γ signaling. In addition, NOTCH-activation causes silencing of the entire *HLA-class II* locus via epigenetic regulation of the transcriptional co-activator CIITA. Notably, while NOTCH1 and NOTCH2 govern similar transcriptional programs, disease-specific differences in their expression levels can favor paralog-specific selection. Importantly, NOTCH-ICD also strongly down-regulates the expression of CD19, possibly limiting the effectiveness of immune-therapies. These NOTCH-mediated immune escape mechanisms are associated with the expansion of exhausted CD8⁺ T cells in vivo.

The landscape of somatic mutations present in malignant cells from patients with B-cell lymphoma has been described in several pivotal sequencing studies[1–3]. In CLL, similar studies have identified more than 40 recurrent mutations mostly affecting oncogenes (*NOTCH1*, Wnt-signaling), tumor suppressors (*TP53, ATM*), and genes involved in RNA-processing (*SF3B1, XPO1, RPS15*)[4–7]. While the prognostic significance is known for some of these mutations, their specific contributions to the pathogenesis of the disease remains largely unknown. Attempts to

[1]Wellcome/MRC Cambridge Stem Cell Institute, Jeffrey Cheah Biomedical Centre, University of Cambridge, Cambridge CB2 0AW, UK. [2]Department of Haematology, University of Cambridge, Cambridge CB2 0AH, UK. [3]Institut d'Investigacions Biomèdiques August Pi i Sunyer (IDIBAPS), Barcelona, Spain. [4]Pathologisches Institut Universität Würzburg, 97080 Würzburg, Germany. [5]Department of Internal Medicine III, Division of CLL, Ulm University, Ulm, Germany. [6]European Molecular Biology Laboratory (EMBL), Heidelberg, Germany. [7]Molecular Medicine Partnership Unit (MMPU), Heidelberg, Germany. [8]Cancer Research UK Cambridge Institute, University of Cambridge, Cambridge, UK. [9]Department of Medical Oncology and Hematology, University Hospital Zürich and University of Zürich, Zürich, Switzerland. [10]Molecular Therapy in Hematology and Oncology, National Center for Tumor Diseases and German Cancer, Research Centre, Heidelberg, Germany. [11]Department of Medical Sciences, University of Turin, Turin, Italy. [12]Institució Catalana de Recerca i Estudis Avançats (ICREA), Barcelona, Spain. ✉e-mail: ir279@cam.ac.uk

address this experimentally have employed genetically engineered mouse models (GEMMs), most commonly the Eμ-TCL1 mouse model[8], and human cell lines. While the former model proved to be useful to recapitulate disease aspects that can only properly be studied in vivo (e.g., tumor-microenvironment interactions), significant limitations exist which prevent extrapolation of data from mice to human. In addition, the experimental manipulation of primary tumor cells from the Eμ-TCL1-model remains technically challenging and most commonly requires crossing of different GEMMs, which consumes time and resources. In contrast, cell lines are easy to manipulate and provide a sheer unlimited and immediate access to tumor cells. However, because they are commonly obtained from patients with end-stage, refractory diseases, frequently are EBV-positive[9,10] and often have been selected for decades to grow in minimal culture conditions, therefore they can lose the biological identity they are supposed to represent. Vigorous proliferation, absence of spontaneous apoptosis, and aberrant homing in NSG mice are some examples for these discrepancies, limiting the conclusions one can draw from such experiments.

Few studies have used adenovirus vectors or their derivates to genetically manipulate primary CLL cells[11]. However, the lack of integration into the host genome results in only transient expression of a gene-of-interest (GOI) and precludes from studying effects in dividing cells or subsequent use of cells in in vivo studies. Alternative attempts using retro- or lentivirus vectors for gene transfer have been unsuccessful for decades, limited by low transfection efficacy (<1%) and substantial toxicity[12], which made downstream analyses impossible. To overcome these limitations, we have developed a method to effectively infect primary neoplastic human B cells using engineered forms of viral envelopes derived from the same gammaretrovirus family. This method permits gene transfer with high transduction efficacy and minimal toxicity, which allows the functional investigations of genes recurrently mutated in primary malignant B cells. We used patient-derived cells from CLL and MCL, two mature B neoplasms with partially overlapping biological features and clinical behaviors[13] that lack appropriate in vitro or in vivo models that span their clinico-biological spectrum.

We have employed this technique to interrogate the molecular functions of NOTCH1 and NOTCH2, which are two of the most commonly mutated genes in B-cell lymphomas. In CLL and MCL *NOTCH*-mutations are associated with a poor clinical outcome and, in the case of CLL, with a high frequency of Richter's transformation[14–16]. Most *NOTCH1* and *NOTCH2*-activating mutations affect their PEST domain, encoded by exon 34[17,18]. In addition, point mutations in the 3'-UTR have been identified which can cause expression of a truncated protein[5]. Both scenarios result in an abnormally stable NOTCH protein, which continues to require ligand binding to become transcriptionally active. Several groups have employed cell lines to study the biology of NOTCH1 in B-cell malignancies and then associated these findings with data from primary cells[19–22]. While such approaches provided important insights into the role of NOTCH1 in CLL, it often remains unclear whether these findings report a direct consequence of activated NOTCH1 or are a mere correlation. Our method to retrovirally infect primary malignant B cells from patients with CLL and MCL to generate isogenic cells provides a unique opportunity to answer this question. Here we provide evidence of how mutated NOTCH favors immune escape of tumor B cells and we address how CLL cells with trisomy 12 may provide a selective advantage for *NOTCH1*-mutations.

## Results
### Retroviral transduction of FeLV-vectors into primary human tumor B cells permits functional downstream analyses
In the past, transduction of CLL B cells using retroviral or lentiviral vectors has largely been unsuccessful[12]. However, as recently demonstrated, normal tonsillar B cells can effectively be transduced with a virus pseudotyped with a Gibbon Ape Leukemia Virus (GaLV) envelope

to transform cells into cells resembling diffuse large B-cell lymphoma (DLCBL)[23]. To investigate whether this vector also allows for the transduction of malignant B cells, we first assessed the expression of *SLC20A1*, the receptor for GaLV, in normal B cells and primary CLL and MCL cells. The abundance of *SLC20A1* mRNA was similar between normal peripheral blood-derived B cells and malignant B cells (Fig. 1a), suggesting that this vector system could also allow successful viral transduction of CLL and MCL cells. We next established a cell culture system, which is reminiscent of a lymph node environment and induces vigorous proliferation of CLL cells to permit retroviral gene transfer. For this, we stably expressed human CD40L in stroma cells derived from murine bone marrows and lentivirally co-transduced with vectors driving the expression of BAFF and human IL-21 (thereafter named MM1 cells). After 72 h of co-culture, we used spinoculation to transduce proliferating CLL cells, which were then continuously cultured on MM1 cells for 7 days (Fig. 1b). Notably, removal of transduced cells from MM1 cells restored their cell cycle arrest, allowing for the investigation of either cycling or arrested tumor B cells (Supplementary Fig. 1a). We initially tested different viral envelopes recognizing the SLC20A1 receptor: While primary malignant B cells were effectively transduced with a GALV envelope (24% in CLL and 12% in MCL), the Feline Leukemia Virus envelopes (FeLV) infected a higher percentage of CLL and MCL cells with an average efficacy of 37% and 39%, respectively. In contrast, CLL cells were resistant to infections with the Simian Sarcoma Associated Virus envelope, which displayed moderate efficacy to infect primary MCL cells (Fig. 1c). Importantly, our transfection method was not associated with increased cell death: after 7 days, CLL cell viability was >80% (Fig. 1d), similar to non-transduced, previously cryopreserved cells and cultured under identical conditions (Supplementary Fig. 1b).

To test that transduced cells remained functionally intact to allow downstream analyses, cells were transduced with a dominant-negative *TP53* (*p53DD*) and subsequently exposed to the anthracycline doxorubicin for 12 h. *P53DD* significantly mitigated *P21* mRNA transcription induced by doxorubicin in cells carrying wild-type *TP53*, but significantly less so in p53-deficient cells (Fig. 1e). Similarly, *p53DD* reduced Fludarabine-induced apoptosis (Fig. 1f). In addition to the interference with a key tumor suppressor, we overexpressed the proto-oncogene *c-MYC* in primary malignant B cells from CLL patients. Expectedly, ectopically expressed *c-MYC* induced a robust proliferative response in CLL cells, indicated by the increased number of cells transitioning through S-phase (Fig. 1g).

In conclusion, we have established a method to effectively infect primary tumor cells from patients with CLL and MCL with minimal toxicity, allowing to generate isogenic, patient-specific tumor cells which differ only in the GOI.

### NOTCH1 drives proliferation, CD38 expression and enhances B-cell receptor signaling
We next used this method to investigate the role of NOTCH1 in primary CLL cells, which is frequently mutated in approximately 10% of untreated patients[17,24]. To simultaneously account for point mutations, missense and frameshift mutations affecting exon 34 and for less common 3'-UTR *NOTCH1* mutations, we expressed the coding sequence of *NOTCH1*-ICD, but lacking the PEST domain, followed by IRES-GFP from the 5'-LTR-promoter (hereafter named *NOTCH1*[ΔPEST]) in primary CLL cells. Importantly, since *NOTCH1*[ΔPEST] also lacked the ectodomain, activation did not require binding of NOTCH ligands or cleavage of the extracellular domain for trans-activation. Since CLL cells co-express NOTCH1 and NOTCH2[25], this approach also allowed us to investigate the function of NOTCH1 in isolation without simultaneously activating other NOTCH receptors. To test that NOTCH1[ΔPEST] was transcriptionally active, we first assessed the mRNA expression levels of *HES1*, *HEY1* and *DTX1*, bonafide NOTCH-target genes. Compared to empty vector (*EV*)

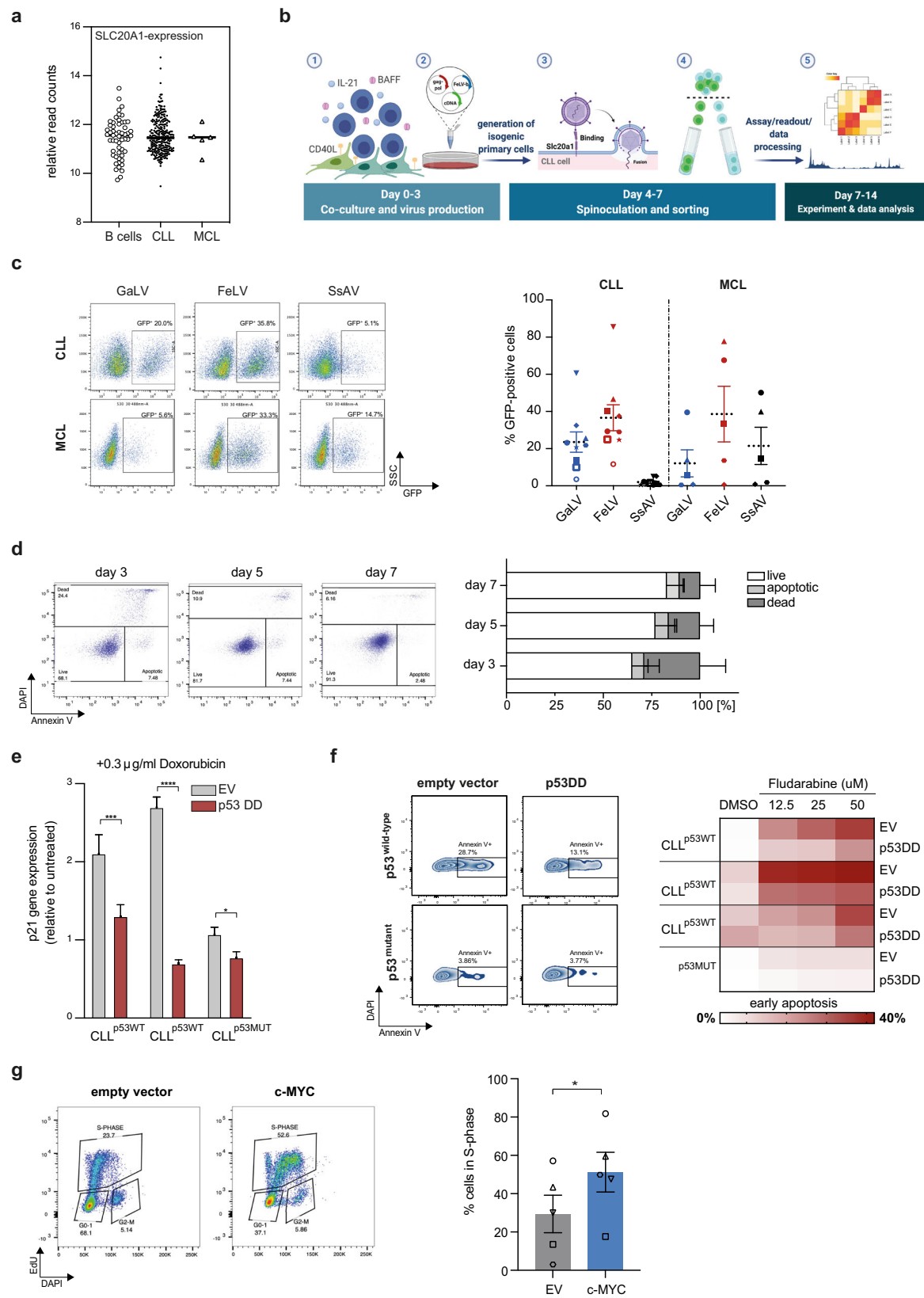

control, NOTCH1$^{\Delta PEST}$ (N1$^{\Delta PEST}$) increased the abundance of *HES1*, *HEY1*, and *DTX1* mRNA by 3.6-, 3.4-, and 3.0-fold, respectively (Fig. 2a). Importantly, similar expression changes of *HES1* and *DTX1* were reported in ligand-activated CLL cells carrying *NOTCH1* mutations[21,22,26,27], indicating that NOTCH1$^{\Delta PEST}$ has a similar activation potential to mutated, endogenous NOTCH1.

Several previous studies have suggested that CLL cells carrying *NOTCH1* mutations have a proliferative advantage compared to wild-type CLL cells[20,27]. To test whether NOTCH1 is involved in cell cycle regulation, we assessed the number of GFP$^+$ cells in S-phase 6 days after transduction. Compared to *EV*-control cells, primary CLL cells expressing *NOTCH1*$^{\Delta PEST}$ consistently showed a higher percentage of

**Fig. 1 | Viral transduction of primary human B cells from CLL/ MCL patients.**
**a** Graph showing *SLC20A1* expression analyzed by RNAseq of naive B cells (n = 55), CLL (n = 283) and MCL (n = 5) primary cells. **b** Schematic representation of the transduction protocol. Primary cells were co-cultured on feeder cells expressing hCD40L, hIL21 and hBAFF for 72 h prior to viral transduction through spinoculation. GFP was used as a marker for successful viral transduction. Created with BioRender.com **c** Transduction-efficiency as determined by GFP expression using three different envelope constructs in CLL and MCL primary cells 72 h post transduction. Representative flow cytometry images of transduction efficacy are shown for each construct on the left. Each symbol refers to an individual patient sample, n = 9 for CLL and n = 5 for MCL. Dotted line = mean ± SEM. **d** Quantification of viable (DAPI⁻ Annexin-V⁻), apoptotic (DAPI⁻ Annexin-V⁺) and dead (DAPI⁺) cells 3-, 5-, and 7 days post transduction (n = 8). Representative flow cytometry images and gating strategy are shown for each time point. **e** qRT-PCR analysis of *p21* mRNA expression in primary GFP-positive CLL cells transduced with an empty vector control (EV) or a

dominant-negative P53 expressing vector (P53DD) after 12 h treatment with doxorubicin, normalized to cells treated with a vehicle control (DMSO). N = 3 individual patient samples, p53WT = wild type for endogenous *TP53*, p53MUT = mutated endogenous *TP53*. The statistical significance was determined using two-tailed paired *t* test calculator (p = 0.0074, 0.0001 and 0.01, respectively). **f** Heatmap showing the percentage of apoptotic (DAPI⁻Annexin-V⁺) CLL cells transduced with either an EV or P53DD after 24 h exposure to Fludarabine. Annexin-V positivity was assessed on GFP only expressing cells. Representative flow cytometry images are shown for each drug concentration on the left (n = 4). **g** S-phase quantification of CLL cells transduced with an empty vector or a MYC expressing vector 6 days post transduction (n = 5). Cells co-cultured on feeder cells were pulsed with Edu for 12 h. Each symbol refers to an individual patient sample. Statistical analysis was done by a two-tailed paired *t*-test (p = 0.036). Cohorts are shown as mean ± SEM. *P < 0.05, **P < 0.01, ***P < 0.001, ****P < 0.0001 and not significant (ns) P > 0.05.

cells going through S-phase (Fig. 2b), associated with weaker staining for Carboxyfluorescein succinimidyl ester (CSFE) (Supplementary Fig. 2a). Accordingly, the fraction of GFP-positive CLL cells continuously increased only in the *NOTCH1^ΔPEST*-transduced cells but not in *EV*-controls, indicating that NOTCH1 positively affects cell cycle progression (Fig. 2c). Importantly, NOTCH1 activation had only minimal pro-apoptotic effects, which were clearly outcompeted by its proliferative advantage (Fig. 2d).

*NOTCH1* mutations have also been associated with surface CD38 expression[28]. In keeping with the observation that CD38 expression is higher in CLL lymph nodes compared to peripheral blood cells[29], co-culture of CLL cells on MM1 cells increased baseline expression of CD38. Under these conditions, NOTCH1^ΔPEST consistently upregulated CD38 expression further, suggesting that its expression is functionally dependent on NOTCH1 activation (Fig. 2e). Of note, NOTCH1^ΔPEST expression did not induce the expression of CD138, *BLIMP1*, or *IRF4* (Supplementary Fig. 2b, c), indicating that CLL B cells did not differentiate into antibody-secreting plasma cells[30]. Other studies have associated NOTCH1 mutations to the expression of CD49d, which equally and independently indicates a poor prognosis. In agreement with a previous study on MEC-1 cells[22], we observed that NOTCH1^ΔPEST induced surface expression of CD49d only in non-trisomy 12 patients, which overall expressed much lower levels of CD49d than trisomy 12 samples (Fig. 2f). Lastly, we assessed B-cell receptor (BCR)-responsiveness of CLL cells transduced with *NOTCH1^ΔPEST* or *EV*. Anti-IgM induced a stronger calcium-flux in cells expressing *NOTCH1^ΔPEST* compared to *EV* cells (Fig. 2g), supporting a recent study which demonstrated a collaboration between NOTCH- and BCR-signaling[26]. In conclusion, retrovirally expressed *NOTCH1^ΔPEST* has biological activities similar to mutated, full-length NOTCH1 and recapitulates functions previously attributed to NOTCH-activation in CLL.

## Patients with trisomy 12 or del13q present a common NOTCH1 transcriptome

To define the global transcriptional program controlled by NOTCH1 and contributing to these phenotypic and proliferative effects, we performed RNAseq on 13 primary CLL samples, either transduced with an *EV*-control or *NOTCH1^ΔPEST*. Only patients who had a deletion of chromosome 13q (del13q) or carried an additional copy of chromosome 12 (tri12), assessed by conventional FISH analyses, were included. Pairwise analysis of all 13 patient samples identified 1636 differentially expressed genes of which 979 were upregulated and 657 were downregulated (applying a cut-off of Log2FC > 0.5, present in at least half of all samples) (Fig. 3a). Gene Set Enrichment Analysis (GSEA) of these differentially expressed (DE) genes identified gene clusters in canonical Notch-signaling (e.g., *HES4*, *SEMA7A*, *CD300A*, and *DTX1*), B-cell activation/ BCR-signaling (e.g., *FYN*, *BLNK*, and *CR2)* and MAPK-activation (e.g., *MAPK8* and *MAP2K6*), in keeping with previous reports based on the ectopic expression of NOTCH1 in lymphoma cell

lines[19,20]. Unexpectedly, NOTCH1 repressed genes were strongly enriched in antigen-processing and presentation, predominately belonging to the family of MHC class II genes (Fig. 3b).

While *NOTCH1* mutations have been identified in less than 15% of treatment naive, unselected patients[4,24], this frequency is significantly higher in patients with trisomy 12, in which *NOTCH1* mutations can be found in 40–50% of patients[31,32]. Importantly, the presence of *NOTCH1* mutations in tri12-CLL is associated with high rates of transformation into Richter's syndrome[14,33]. The underlying reasons for this peculiar association are unknown, but it suggests that NOTCH1 may regulate distinct, transformation-favoring genes in cells carrying an extra chromosome 12. To address this hypothesis, we separately analyzed the gene expression profiles induced by NOTCH1 in 7 tri12 and 6 del13q patients. Pairwise analysis (applying a cut-off of Log2FC > 0.5 in at least half of all samples and of Log2FC > 0 in all samples) identified 410 NOTCH1-regulated genes in tri12 (268 upregulated/142 downregulated) and 418 genes in del13q (236 upregulated/182 downregulated) patient samples. DE genes in each group were then used to validate the NOTCH1 transcriptome on a cohort of *NOTCH1*-mutated tri12 and del13q patients[34] and showed a significant enrichment in the respected genotype (Fig. 3c). The comparison of NOTCH1-induced genes did not identify a distinct expression profile in tri12 compared to del13q cells, as shown by the correlation of the respective logFCs in both sets of deregulated genes (Fig. 3d). Interestingly, the slope of the tri12-specific genes was clearly smaller than that of del13q-specific genes, suggesting that NOTCH1^ΔPEST induced a higher amplitude of gene activation/deactivation in the tri12 genetic context (Fig. 3d, black vs gray line). Furthermore, this analysis recognized 130 genes that were deregulated at similar levels by NOTCH1^ΔPEST in all 13 samples, independent of their genetic background (Fig. 3d, red line and Supplementary Data 1). Subjecting this gene set to GSEA identified transcriptional changes in gene clusters involved in signaling, stress-response, and antigen-presentation (Supplementary Fig. 3a, b). Notably, despite promoting proliferation of CLL cells, we did not observe an increased expression of *MYC* or cell cycle genes in *NOTCH1^ΔPEST* transduced cells (Supplementary Fig. 3c).

In addition to a clear transcriptional modulation, we next assessed whether NOTCH1^ΔPEST was also able to induce epigenetic programming at the level of chromatin regulation. For this, we performed ChIP-seq for histone H3 lysine 27 acetylation (H3K27ac), a bonafide mark for active regulatory elements[35], in five paired CLL samples (expressing either *EV*-control or *NOTCH1^ΔPEST*). Unsupervised principal component analysis revealed that the first components of the chromatin activation variability were patient-specific. However, the fifth component, which explains 3.6% of the total variability, remarkably separated controls from NOTCH1^ΔPEST expressing samples, regardless of their genetic background (Fig. 3e). This NOTCH1^ΔPEST-associated signature was composed of 587 H3K27ac peaks (Supplementary Data 2). Of them, 422 peaks were located at active chromatin states of CLL reference epigenome samples[36] and at the promoter or gene body of an

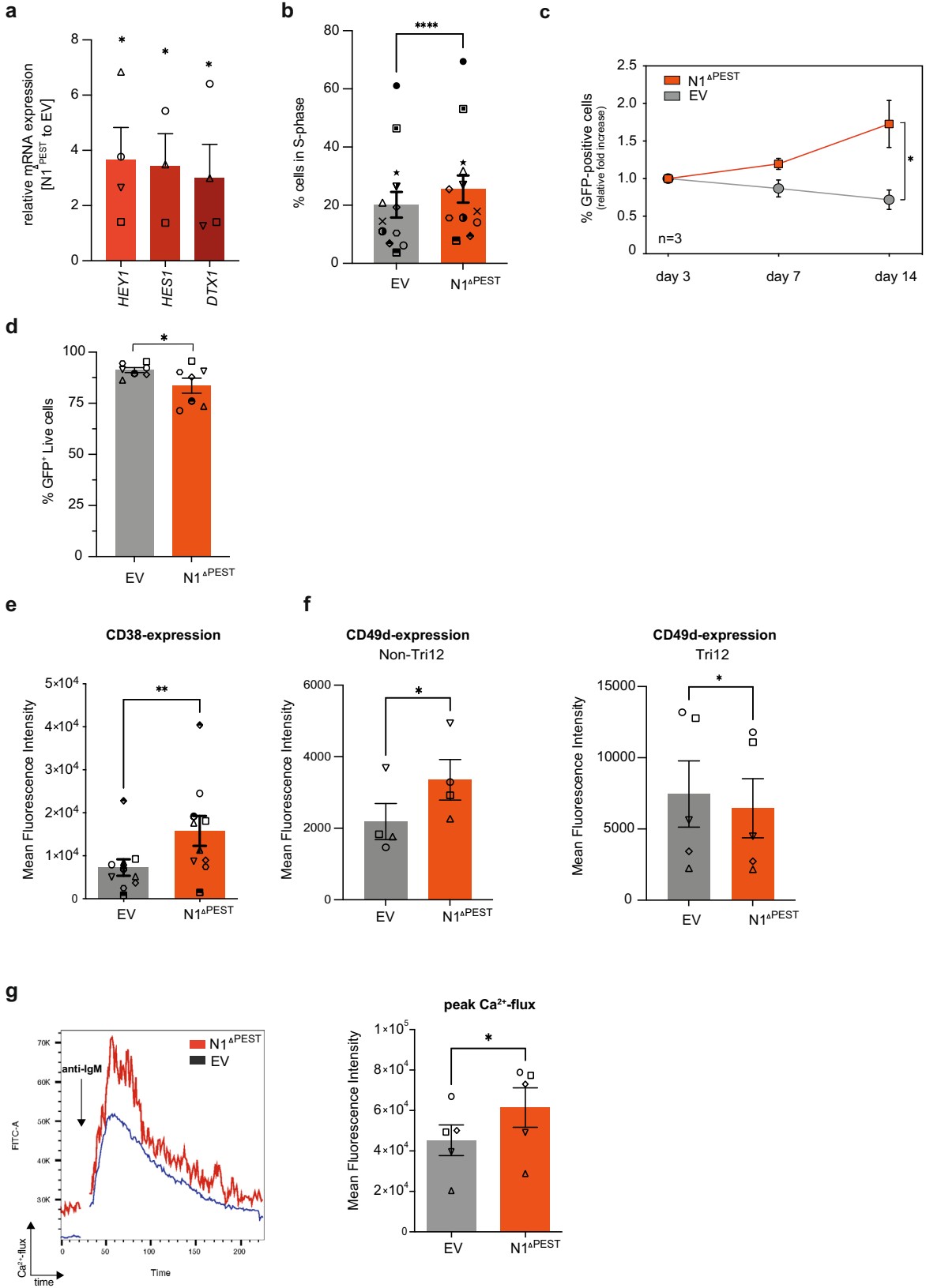

annotated gene (see materials and methods). Furthermore, at those differentially acetylated peaks we observed a consistent increase or decrease in H3K27ac signal in all cases with NOTCH1$^{\Delta PEST}$ regardless of the cytogenetic background (Fig. 3f). Consistently with the RNAseq data, we identified chromatin activation at several NOTCH1 target genes such as *HES1* (Fig. 3g). These data demonstrate that NOTCH1

mutations not only drive transcriptional changes but also induce an aberrant epigenetic programing of CLL cells.

Collectively, these results indicate that NOTCH1 activation positively regulates gene expression important for B-cell activation while simultaneously repressing genes required for antigen presentation. These effects were not qualitatively different in tri12 cells, but here

**Fig. 2 | NOTCH1$^{\Delta PEST}$ recapitulates biological features of mutated full-length NOTCH1. a** qRT-PCR analysis of canonical NOTCH1-target genes expression (*HEY1, HES1, DTX1*) in primary CLL cells transduced with *NOTCH1$^{\Delta PEST}$*. Expression was normalized to cells transduced with an empty vector control (*n* = 5). Cells were FACS sorted for GFP expression prior to RNA extraction 5 days post transduction (*p* = 0.04; 0.049 and 0.047, respectively). **b** S-phase quantification of CLL cells transduced with an *empty vector* (gray) or *NOTCH1$^{\Delta PEST}$* (red) (*n* = 13). Cells were pulsed with Edu for 12 h. Cell cycle analysis was restricted to GFP-positive cells. **c** Growth curve of CLL cells transduced with an *empty vector* (gray) or *NOTCH1$^{\Delta PEST}$* (orange) quantified as GFP expression 3-, 7- and 14-days post transduction. The ratio of GFP-positive cells compared to day 3 is shown (*n* = 3); (*p* = 0.04). **d** Quantification of viable (GFP$^+$DAPI$^-$AnnexinV$^-$) cells 5 days post transduction (*n* = 7); (*p* = 0.049). **e** CD38 expression on CLL cells transduced with an *empty vector* (gray) or *NOTCH1$^{\Delta PEST}$* (orange) 5 days post transduction (*n* = 10), assessed by flow cytometry (*p* = 0.001). **f** CD49d expression on *del13q* (*n* = 4, *left panel*) or Tri12 primary CLL cells (*n* = 5, *right panel*) transduced with an *empty vector* (gray) or *NOTCH1$^{\Delta PEST}$* (orange) 5 days post transduction (*p* = 0.023 for del13q and *p* = 0.025 *for Tri12*). **g** Quantification of peak Ca$^{2+}$-flux in interval time in response to anti-IgM stimulation of CLL cells expressing NOTCH1$^{\Delta PEST}$ identified by GFP expression (*n* = 5). Interval times were automatically determined by kinetic analysis using the FlowJo software. Representative Ca$^{2+}$-flux dot plots overlap of a single patient is shown on the left showing the induction of Ca$^{2+}$-flux following IgM stimulation (*p* = 0.014). Each symbol refers to an individual patient sample. Error bars are shown as mean ± SEM except for panel c, which shows ±SD. The statistical significance was determined by two-tailed paired *t* tests. *$P < 0.05$, **$P < 0.01$, ***$P < 0.001$, ****$P < 0.0001$ and not significant (ns) $P > 0.05$.

NOTCH1 effects seemed to be more enhanced compared to del13q cells.

## NOTCH1 represses MHC class II genes via downregulation of *CIITA*

Our RNAseq analyses indicated that NOTCH1 is associated with reduced expression of genes important for antigen-presentation, including *HLA-DM, -DR, -DP,* and *-DQ* (Supplementary Data 1), suggesting silencing of the MHC class II locus on chromosome 6. Indeed, H3K27ac ChIP-seq analysis confirmed that this gene repression was due to epigenetic silencing of the entire *HLA*-locus (Fig. 4a) and demonstrated that, besides gene activating functions, NOTCH1 can also induce repressive effects on transcription. Assessment of surface HLA-DR expression on an additional 12 primary CLL samples confirmed that NOTCH1 activation is consistently associated with downregulated *HLA-class II* genes (Fig. 4b). We could not recapitulate this effect in CLL cell lines (Supplementary Fig. 4a). To provide further evidence for the gene repressive functions of NOTCH1, we cultured CLL cells from 4 donors with endogenous exon 34 mutations of *NOTCH1* under identical conditions on MM1-stroma cells in the presence or absence of γ-secretase inhibitors (GSI) to block ligand-mediated activation of NOTCH1. As shown by us and others, stroma cells express NOTCH ligands[37,38], which can trigger activation of the Notch pathway. Blockade of NOTCH-activation by GSI treatment induced a significant downregulation of *HES1* and *HEY1* (Supplementary Fig. 4b) and upregulation of *HLA-DR* in *NOTCH1*-mutated CLL, supporting that NOTCH1 can repress the expression of *HLA*-genes (Fig. 4c).

The ubiquitous repression of the entire *HLA-class II* locus suggested that NOTCH1-activity could affect the expression of the class II transactivator (*CIITA*) in CLL cells, which is a master regulator for *HLA class II* genes. CIITA, which itself does not bind to DNA, interacts with multiple transactivating proteins of the MHC class II enhanceosome and regulates gene expression through multiple mechanisms, including recruitment of transcription factor IID, phosphorylation of RNA polymerase II, and histone modification (reviewed in ref. 39). Consistent with our hypothesis, *CIITA*-RNA levels were significantly downregulated in primary CLL cells transduced with *NOTCH1$^{\Delta PEST}$* (Fig. 4d) and H3K27ac ChIP-seq showed silencing of the *CIITA* locus on chromosome 16 in all samples (Fig. 4e). In conclusion, our data provide evidence that NOTCH1 down-regulates *HLA-class II* genes via transcriptional suppression of CIITA.

The downregulation of *MHC class II* genes provides an immune escape for cancer cells by reducing their immunogenicity. The clinical significance of the downregulated or absent HLA-class II expression in B-cell lymphoma is illustrated by its negative impact on the prognosis of patients with DLBCL and primary mediastinal B-cell lymphoma (PMBCL)[40,41]. To test whether CIITA-dependent downregulation of *MHC class II* genes was prognostically important also for CLL, we analyzed whether CIITA-RNA levels predicted the time to first treatment in a cohort of 266 treatment naive patients from the International Cancer Genome Consortium (ICGC)[5,42]. Dividing patients into either *CIITA* low or high expresser, based on the overall *CIITA* mRNA abundance, we discovered that those patients with low *CIITA* levels had a significantly more active disease and required treatment sooner (Fig. 4f). Importantly, *NOTCH1* exon 34 mutations were twice as common in the *CIITA* low expresser group compared to high expresser (Fisher *t*-test, *p* = 0.032) (Fig. 4g).

In addition, we assessed the significance of *CIITA* expression in a cohort of pre-treated CLL patients from the CLL-2H study[43]. To also consider NOTCH1 activation in the absence of exon 34 mutations[19], we first stratified 337 treatment-naive patients based on the expression of canonical target genes *HES1/2, HEY1/2,* and *CIITA* expression (median high vs. median low). This analysis identified a group of patients with high and low expression of *CIITA* in both NOTCH1-activated and non-activated groups, defining 4 patient cohorts (Supplementary Fig. 4c, d). We applied these expression thresholds to gene expression data generated from PBMCs in a cohort of fludarabine-resistant CLL treated in the CLL-2H study. These analyses indicated that high expression of canonical NOTCH-target genes was not per se associated with an unfavorable prognosis, but significantly impacted on the overall survival in combination with low levels of *CIITA* expression (Fig. 4h). Importantly, within the *CIITA$^{low}$* expresser, *NOTCH1* mutations were significantly more frequent in the *NOTCH$^{high}$* versus *NOTCH$^{low}$* group (Fig. 4i).

Collectively, these data demonstrate that low levels of *CIITA* are associated with a more aggressive disease, in particular for patients with activated Notch-signaling. Furthermore, this analysis also suggests that NOTCH1 can be activated in the absence of *NOTCH1*-mutations, as previously reported[19].

## NOTCH1 up-regulates PD-L1 and impairs T-cell activation

NOTCH1-dependent suppression of *CIITA* and further downstream *HLA-class II* genes indicated a mechanism for an escape from immune surveillance. To provide further evidence for NOTCH1-mediated phenotypic changes which could impinge on T-cell activation, we performed quantitative proteomic analysis on primary CLL cells transduced with *NOTCH1$^{\Delta PEST}$* or *EV*-control. For the simultaneous analysis of both groups of samples, we applied Tandem Mass Tags (TMT-6plex), which allows for the quantitative comparison between replicates and conditions. Total proteomic investigation of 3 patients recognized 7876 unique proteins and as a result of pairwise analysis, we identified 385 differentially regulated proteins (average Log2FC > 0.5 and all 3 patients with Log2FC > 0; Fig. 5a and Supplementary Data 3). Proteins expressed at a higher level included positive controls such as NOTCH1 and CD38. Interestingly, NOTCH1$^{\Delta PEST}$ increased the expression of CD27 and CD274 (PD-L1) in CLL cells, which both can impair T-cell activation. Assessment of PD-L1 expression on additional 20 CLL patients, transfected with *NOTCH1$^{\Delta PEST}$* or *EV*, invariably showed an upregulation of PD-L1 through activated NOTCH1. Notably, we were unable to recapitulate this phenotype in the CLL cell lines MEC-1 and Hg-3 (Fig. 5b), further emphasizing the limitations inherent to studies with cell lines. To demonstrate

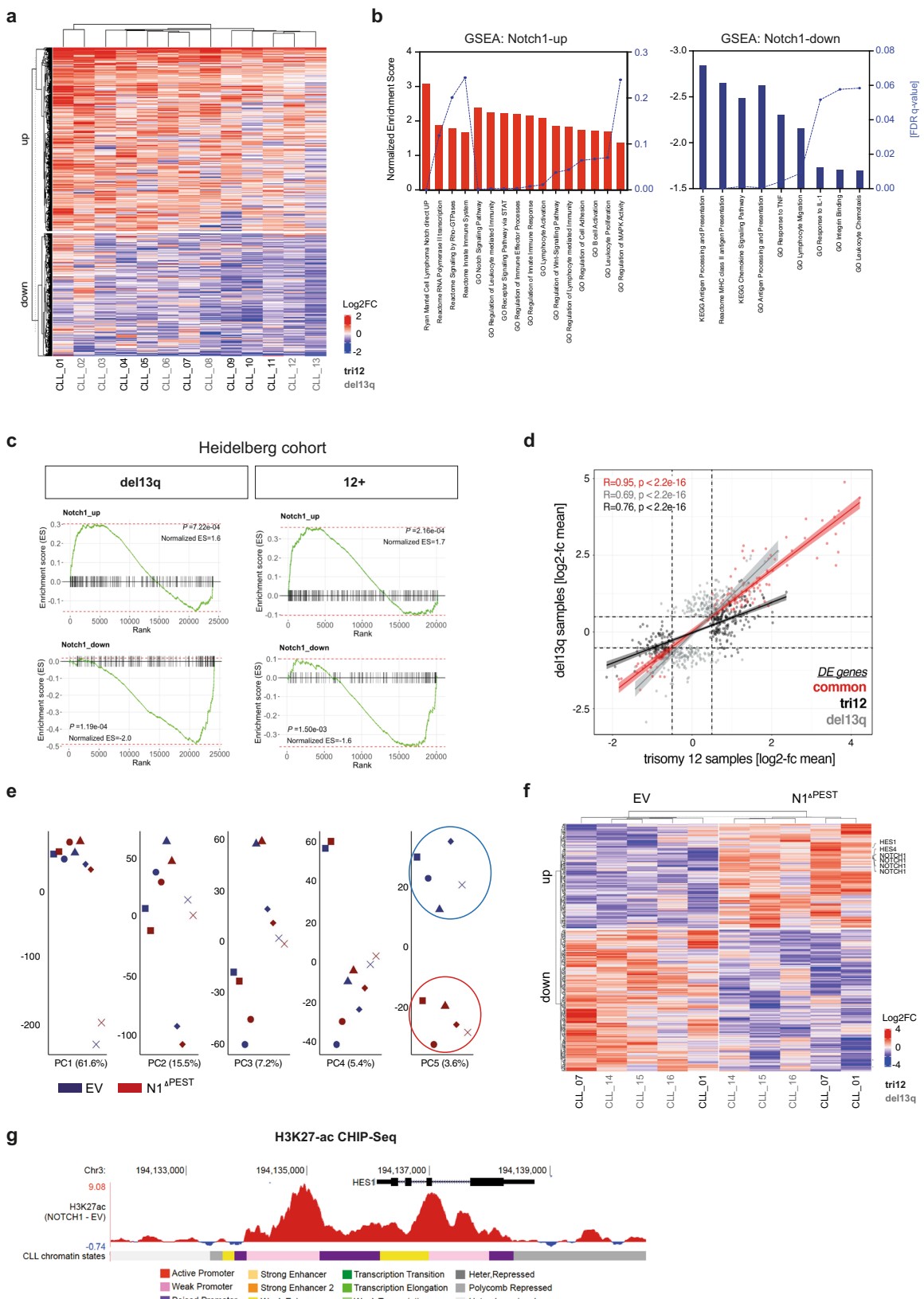

that NOTCH1$^{\Delta PEST}$ functions similarly to ligand-activated, mutated NOTCH1, we performed a reverse experiment by treating *NOTCH1*-mutated CLL with γ-secretase inhibitors (GSI) to block NOTCH-activation. GSI treatment caused a significant downregulation of PD-L1 in primary *NOTCH1*-mutated CLL (Fig. 5c). Importantly, we observed that the constitutive expression of PD-L1 on quiescent cells was minimal, but significantly upregulated on activated, cycling CLL cells (Supplementary Fig. 5a), in keeping with a report showing strong expression of PD-L1 on CLL cells in proliferative centers in lymph nodes[44]. To test whether CLL cells recently egressed from lymph nodes had higher expression levels of PD-L1, we assessed its expression on peripheral blood cells expressing CXCR4$^{dim}$/CD5$^{bright}$ (ref. 45).

**Fig. 3 | Gene expression profiles of NOTCH1[ΔPEST] transduced CLL cells. a** Heatmap DE genes following NOTCH1[ΔPEST] overexpression in CLL cells (n = 13). Libraries were generated from mRNA isolated from FACS sorted GFP[+] cells 4 days post infection. **b** Gene ontology (GO) and Gene set enrichment analysis (GSEA) results of NOTCH1[ΔPEST] DE genes identified by RNAseq. Upregulated (left) and downregulated (right) gene-sets are indicated by bars. Normalized Enrichment Scores are shown on the left black Y-axis, FDR q values on the right Y-axis. **c** DE genes identified by RNAseq were validated on a cohort of NOTCH1-mutated patients with del13q (left panel) or trisomy 12 (right panel). **d** Scatter plot showing the Log2FC mean of DE genes in tri12 (n = 7, light gray dots, x-axis) vs del13q (n = 6, black dots y-axis) following NOTCH1[ΔPEST] overexpression. Red dots represent commonly DE genes between the two different groups. Error bands indicate the 95% percentile of the mean. **e** Unsupervised principal component analysis of the H3K27ac ChIP-seq profiles of n = 5 paired CLL primary cases transfected with empty vector (EV) or NOTCH1[ΔPEST] (N1[ΔPEST]). 43,300 independent genomic regions were analyzed to generate the PCAs. **f** Heatmap shows VST normalized H3K27ac signals for those peaks identified in principal component 5, associated with promoters or gene bodies (n = 422 peaks). **g** Example targeting HES1 gene, identified to have an increase H3K27ac signal in NOTCH1[ΔPest] expressing samples. The mean value of the subtracted signal calculated for each individual paired sample per 1 bp is shown. One representative analysis out of 5 is shown.

Unexpectedly, we did not find a different expression of PD-L1 compared to CXCR4[bright]/CD5[dim] cells, which was overall very low (Supplementary Fig. 5b). These data strongly suggest that the upregulation of PD-L1 in proliferative centers is short-lived.

PD-L1 is also post-transcriptionally regulated through cyclinD-Cdk4 activity, causing cell cycle dependent oscillations of PDL-L1 expression with a peak expression during M- and early G1-phase[46]. Since NOTCH1[ΔPEST] provided a modest, but significant proliferative advantage for primary CLL cells (Fig. 2b, c), we hypothesized that the increased expression of PD-L1 could be attributed to an increased proliferation, rather than being a specific NOTCH1-response. To address this, we assessed the expression of PD-L1 on cycling CLL cells. In keeping with cell cycle modulated expression of PD-L1, we observed a significant downregulation on CLL cells going through S-phase, which recovered with entry into G2/M-phase. Notably, the expression of PD-L1 was consistently increased in NOTCH1[ΔPEST]-transduced CLL cells compared to EV-controls (Fig. 5d), indicating that the NOTCH1-induced expression of PD-L1 is not dependent on cell proliferation. In addition, although retroviral expression of MYC caused rigorous proliferation of primary CLL cells (Fig. 1g), c-MYC did not affect PD-L1 expression (Fig. 5e), providing further evidence for a cell cycle and MYC- independent regulation of PD-L1 by NOTCH1.

Besides the post-transcriptional regulation of PD-L1, its expression is induced by a variety of pro-inflammatory cytokines, of which interferons are strong inducers. Importantly, CLL cells can produce and secrete IFN-γ, which provides an anti-apoptotic signal through autocrine stimulation[47]. To investigate whether this feedback loop was affected by NOTCH1 activation, we analyzed the mRNA expression of several pro-inflammatory genes in NOTCH1[ΔPEST] transduced cells. While the expression of most of these investigated genes was unaffected by NOTCH1 activation, it clearly upregulated IFN-γ and its putative receptor (Fig. 5f and Supplementary Fig. 5c), suggesting a contribution of autocrine secreted IFN-γ to NOTCH1-mediated expression of PD-L1. In support of this hypothesis, both PD-L1 and IFNGR1 show a distinct H3K27ac peak at the promoter regions in the NOTCH1[ΔPEST] samples compared with the controls (Supplementary Fig. 5d). Indeed, CLL cells cultured in the presence of an antibody blocking the IFN-γ receptor showed a significantly downregulation of PD-L1 (Fig. 5g), which was still significantly higher than PD-L1 levels of EV-control cells, indicating that this pathway only partly contributes to the NOTCH1-mediated upregulation of PD-L1.

To demonstrate that the upregulation of PD-L1 by NOTCH1 was functionally important and could indeed impair T-cell activation, we co-cultured primary CLL cells, either transfected with EV or NOTCH1[ΔPEST], with Jurkat T cells, expressing luciferase under the control of the NFAT responsive element (Fig. 5h). Under these conditions, T-cell activation was strictly dependent on the presence of the anti-CD19/ anti-CD3 bi-specific antibody Blinatumomab. Unexpectedly, surface expression of CD19 was significantly downregulated by activated NOTCH1 (Fig. 5i) and NOTCH1[ΔPEST] transduced primary CLL cells mitigated the activation of Jurkat T cells. Notably, blockade of PD-L1 with durvalumab antagonized the NOTCH1-dependent, inhibitory effects on Jurkat T cells (Fig. 5j), indicating that PD-L1 upregulation by NOTCH1 and not reduced CD19 expression impaired T-cell activation in this experiment.

In conclusion, these results demonstrate that NOTCH1 signaling promotes escape from immune surveillance through transcriptional regulation of HLA-class II genes and PD-L1.

## NOTCH1 and NOTCH2 govern similar transcriptional programs in MCL

Contrary to CLL, NOTCH2 mutations have also been identified in MCL with similar hotspots in the PEST domain and frequency than NOTCH1 mutations. Both NOTCH mutations appear to be mutually exclusive in MCL and are associated with a more aggressive clinical course[18]. To assess whether the disease-specific occurrences of NOTCH-mutations were associated with differences in expression of either NOTCH1 or NOTCH2, we analyzed expression data of primary CLL (n = 54 cases) and MCL (n = 54 cases, 30 conventional MCL, and 24 non-nodal MCL)[48]. Notably, expression levels of NOTCH1 were only slightly higher in CLL than in MCL, whereas NOTCH2 was significantly more abundant in MCL. These differences were not attributed to higher frequencies in NOTCH1 or NOTCH2 mutations, respectively (Fig. 6a).

Whether NOTCH1 and NOTCH2 paralogs exert similar, different, or even contrary biological functions remains a controversial question in the field and is cell and context dependent. Notably, NOTCH1-ICD and NOTCH2-ICD are structurally related but are only 59% homologous and have been shown to regulate distinct transcriptional programs in myeloid cells[49], possibly regulated through multiple post-translational modifications (reviewed in ref. [50]). Therefore, we overexpressed NOTCH1[ΔPEST] or a similarly truncated NOTCH2-ICD (thereafter named NOTCH2[ΔPEST]) to define whether their transcriptomes differ in primary MCL cells. Unsupervised principal component analysis of RNAseq data from 3 paired MCL samples indicated that the third component of the PC, explaining 6.5% of the total variability, clearly separated controls from NOTCH1[ΔPEST]/ NOTCH2[ΔPEST] expressing cells, regardless of their patient-specific background (Fig. 6b). However, none of the principal components showed a separation between NOTCH1[ΔPEST] and NOTCH2[ΔPEST] samples, indicating they induce similar transcriptional changes. To further explore differences in gene expression induced by NOTCH1[ΔPEST] and NOTCH2[ΔPEST] we analyzed the 500 genes that showed the highest variable expression across all samples, and observed their expression was highly correlated in all three paired samples (Fig. 6c). This further suggested that NOTCH1/2 ICD regulate similar genes in MCL. Further characterization of the expression signature commonly driven by NOTCH1[ΔPEST] or NOTCH2[ΔPEST] identified 78 differentially expressed genes (57 up- and 21 downregulated) between control or NOTCH1/NOTCH2 MCL samples (Deseq2 analysis, adjusted p value < 0.05; Fig. 6d). Importantly, besides upregulated bonafide NOTCH-target genes (HES1, HEY), NOTCH2 also downregulated CIITA and HLA-class II genes in MCL. Reduced expression of CIITA and HLA-DR in a NOTCH[ΔPEST] expressing MCL-cell line (Fig. 6e) or primary MCL cells (Fig. 6f and

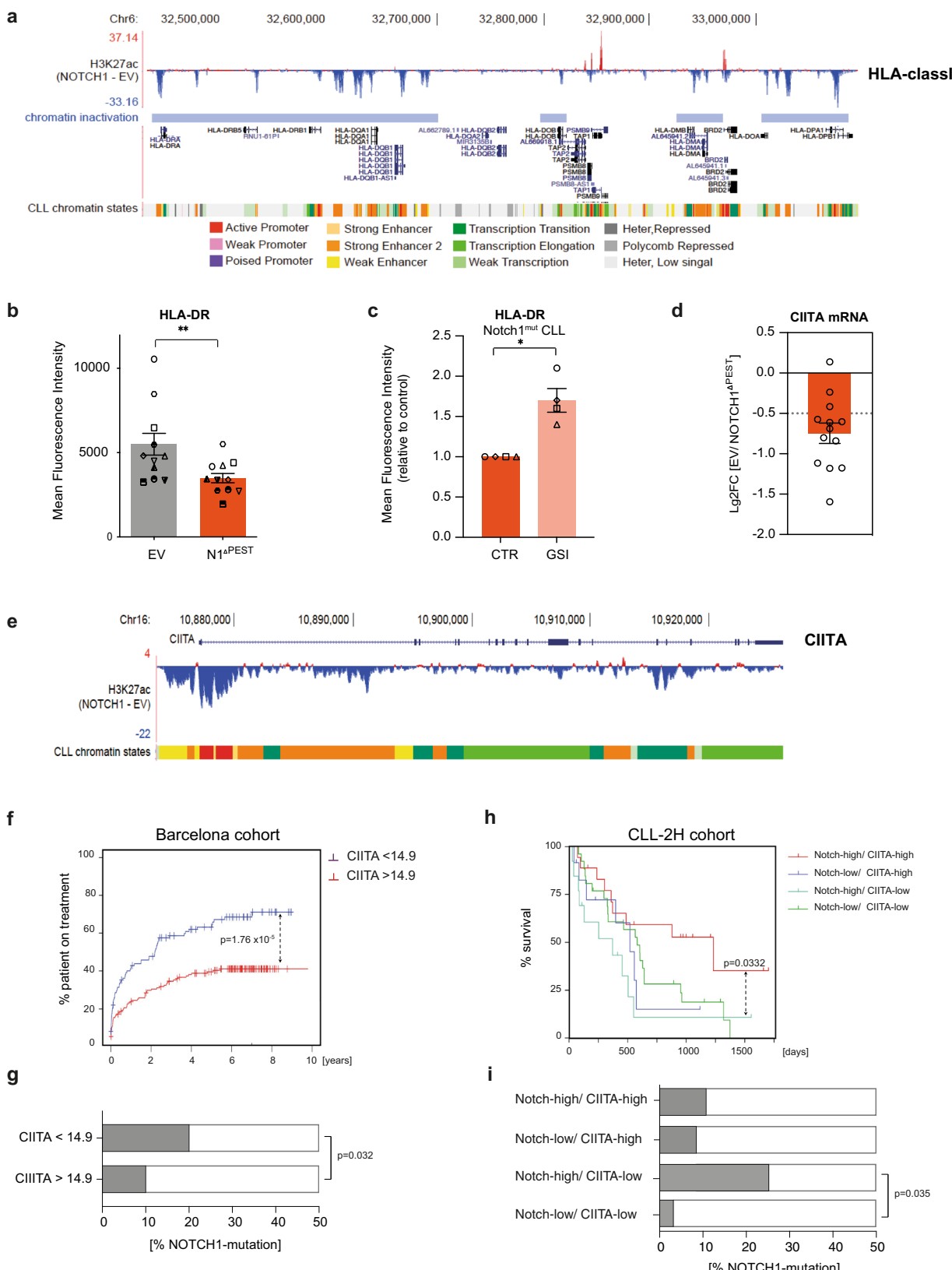

In summary, these data indicate that both, NOTCH1 and NOTCH2, drive a similar immunosuppressive transcriptional program in MCL, alike mutated NOTCH1 does in CLL. The occurrence of *NOTCH2* mutations in MCL, but not CLL, may be related to higher baseline expression levels and clonal selection of mutated *NOTCH2*.

Supplementary Fig. 6a) was confirmed by assessment of protein levels. In addition, activated NOTCH1 and NOTCH2 upregulated PD-L1 expression in primary MCL cells, but not in Jeko-1 cells, further underlying the limitations of studies in cell lines (Fig. 6g and Supplementary Fig. 6b).

**Fig. 4 | NOTCH1$^{\Delta PEST}$ represses MHC class II genes in CLL. a** H3K27ac ChIP-seq profile of the entire MHC class II locus following NOTCH1$^{\Delta PEST}$ overexpression. The peaks represent the mean of the ratio of values obtained from CLL cells transduced with *NOTCH1$^{\Delta PEST}$* or with an *empty vector* control (n = 5). Each gene and loci locations are shown. **b** HLA-DR expression quantified by flow cytometry of transduced CLL cells (n = 12). HLA-DR expression was analyzed on GFP-positive cells 5 days post transduction. Each symbol refers to an individual patient sample (p = 0.002). **c** Quantification of HLA-DR expression on *NOTCH1*-mutated CLL cells treated for 48 h with a γ-secretase inhibitor (10 μM) or DMSO as control. Relative expression compared to empty vector control is shown (n = 4). Each symbol refers to an individual patient sample. (p = 0.017). **d** CIITA mRNA expression analyzed by RNAseq of n = 13 CLL patients. **e** H3K27ac ChIP-seq profile at *CIITA* locus. The peaks represent the mean of the ratio of values obtained from CLL cells transduced with

*NOTCH1$^{\Delta PEST}$* compared to an *empty vector* control (n = 5). Chromatin states are color-coded, corresponding to the legend in panel **a**. **f** Time to treatment (TTT) curve of a cohort of patients classified as expressing high (red line, CIITA > 14.9, n = 156) or low (blue line < 14.9, n = 110) CIITA level. **g** Relative percentage of distribution of NOTCH1-mutated patients in the two sub-groups. Statistical significance was assessed by a Fisher *t*-test. **h** Survival of a cohort of patients with a high *NOTCH1* gene signature identified by high expression of (*HES1/2, HEY1/2*) and further classified by high or low *CIITA* expression (upper panel). **i** Relative percentage of distribution of *NOTCH1* mutated patients in the 4 sub-groups. Statistical analysis was done by a Mann–Whitney test. Error bars are shown as mean ± SEM. Statistical analysis was done by two-tailed paired *t*-tests or log-rank tests (**f, h**). *P < 0.05, **P < 0.01, ***P < 0.001, ****P < 0.0001 and not significant (ns) P > 0.05.

## NOTCH1 activation in CLL cells favors expansion of CD4$^+$ cells in vivo

Our data demonstrated that PD-L1 expression was significantly upregulated in cycling CLL cells, further enhanced through NOTCH1 activation. To provide in vivo evidence supporting this finding, unselected lymph node specimens from CLL/SLL patients were retrieved from the files of the Institute of Pathology, Würzburg, Germany and stained for NOTCH1. We found nuclear expression of NOTCH1 in 12% of all samples (Fig. 7a). Although genomic data for these samples were not available, this frequency is expected based on the reported occurrence of *NOTCH1* mutations in an unselected patient cohort[24]. For multiparameter analysis of the lymph node microenvironment, we employed imaging-mass cytometry (IMC) and applied a panel of isotype-labeled antibodies against B- and T-cell epitopes on paraffin-embedded tissues (Fig. 7b). Following cell segmentation using the CellProfiler software we identified areas with high Ki67 signal using HistoCat software[51] to specifically gate on proliferative centers (PCs). IMC-single cell data identified a high percentage of T cells present in PCs (Fig. 7c and Supplementary Fig. 7a). Analysis of PD-L1 expression on CD19$^+$ cells revealed a stronger signal in PC areas compared to non-PC areas in all samples, in agreement with published data[44]. Further analyses were restricted to B cells in PCs and showed that nuclear NOTCH1 expression was associated with higher PD-L1 and Ki67 signals, compared to NOTCH1 negative samples (Fig. 7d). Assessment of T cells in PCs also showed a significantly higher infiltration of CD4$^+$ cells in NOTCH1-positive samples, associated with higher expression of Ki67, suggesting that NOTCH1 expression in CLL cells promotes T-cell expansion. PD-1 levels on CD4$^+$ cells were similar between NOTCH-positive and negative samples (Fig. 7e). Similar to CD4$^+$ cells, CD8$^+$ cells were also more abundant in PC of NOTCH1-positive patient samples and they also expressed higher levels of PD-1 (Fig. 7f) in contrast to CD4$^+$ cells. These results confirmed our in vitro data of NOTCH1-mediated regulation of PD-L1 and indicated that NOTCH1 supports proliferation of CD4$^+$ and CD8$^+$ cells, with only the latter having a more exhausted phenotype.

To provide further experimental evidence corroborating these findings, we injected *NOD.Cg-Prkcd $^{scid}$Il2rd $^{tm1Wjl}$/Szj* (NSG) mice intraperitoneally with isogeneic primary CLL cells, carrying either *NOTCH1$^{\Delta PEST}$* or *EV*-control. Prior to this, autologous T cells were isolated with anti-CD3 beads, cultured for 7 days and then co-injected at a ratio of 20:1 (CLL:T cell) (Fig. 7g). Assessment of engrafted CLL cells showed a moderate, but not significant, increase in the tumor burden of *NOTCH1*-transduced cells after 3 weeks (Fig. 7h). Importantly, we observed a significant increase of CD4$^+$ T cell in the peritoneal cavity of mice diseased with NOTCH1$^{\Delta PEST}$-expressing CLL cells, demonstrating that NOTCH1 promotes expansion of CD4$^+$ cells (Fig. 7i). This is further supported through clinical data from untreated CLL patients, showing a higher ratio of CD4$^+$/CD8$^+$ cells in the peripheral blood of *NOTCH1* mutated patients compared to NOTCH-wild-type patients (Fig. 7j).

In summary, our data describe immune escape mechanisms governed by mutated *NOTCH* in mature B-cell malignancies, mediated

by increased PD-L1 expression and downregulation of MHC class II genes (Fig. 8).

## Discussion

Numerous sequencing studies have identified many mutations recurrently found in malignant B cells from CLL and MCL patients[4,24,52]. To translate this knowledge into patient care, functional studies are needed to understand the mechanisms governed by these mutations and to identify downstream effects amenable for therapeutic interventions. Here we provide a method to functionally interrogate gene mutations in primary human malignant B cells. For a disease such as MCL or for studying tumor cells with structural chromosomal abnormalities, for which no animal models exist, this method is indeed a unique opportunity to decipher the underlying disease biology.

We applied this technique to address the question why CLL and MCL patients carrying *NOTCH* mutations have a dismal prognosis. Previous studies had approached this question through investigations in cell lines, commonly derived from therapy-resistant patients. Undoubtably, these studies have made significant contributions to our understanding of the NOTCH1 biology and described that it promotes proliferation, BCR-signaling, MAPK-signaling, and chemotaxis in CLL cells[19–21,26,53]. Our studies with primary malignant B cells indeed confirmed these findings, but also identified yet unappreciated roles of NOTCH1. Concerns that expression of NOTCH-ICD exerts 'supra-pathophysiological' effects of NOTCH1, not seen with PEST domain deleted endogenous NOTCH1, were overcome by our quantitative analyses of target gene expression, reverse regulation of PD-L1 and HLA-DR by GSI treatment of *NOTCH1*-mutated CLL and numerous in vivo analyses.

Since most *NOTCH* mutations in CLL and MCL do not affect protein binding to DNA but instead impair its proteasomal degradation by truncating the PEST domain[5,24,52,54], several aspects need to be considered to fully comprehend when, where and how NOTCH signaling drives disease progression. The preserved DNA-binding functions of mutated NOTCH suggest that it regulates the expression of identical genes than wild-type NOTCH and that disease-promoting events are rather caused through secondary effects attributed to signal persistence. Thus, mutated NOTCH still requires binding of NOTCH ligands, expressed in trans, for signaling. Unfortunately, in CLL as well as in MCL, very limited knowledge exists about the expression of NOTCH ligands and receptors in distinct niches in vivo. Extrapolating from our recent in vitro data, ligand and receptor expression are also likely to be dynamic in vivo and regulated by Notch-signaling itself[38]. Another unknown variable, essential for understanding how *NOTCH* mutations modulate disease biology is the length of time a tumor cell resides in one tissue before migrating to another, which will impact on the activation of NOTCH1 in tumor cells as signaling is cell-contact dependent.

We believe this complexity of NOTCH-signaling is important for understanding the recently reported activation of NOTCH1 in 50% of CLL patients, based on the presence of NOTCH1-ICD protein, although

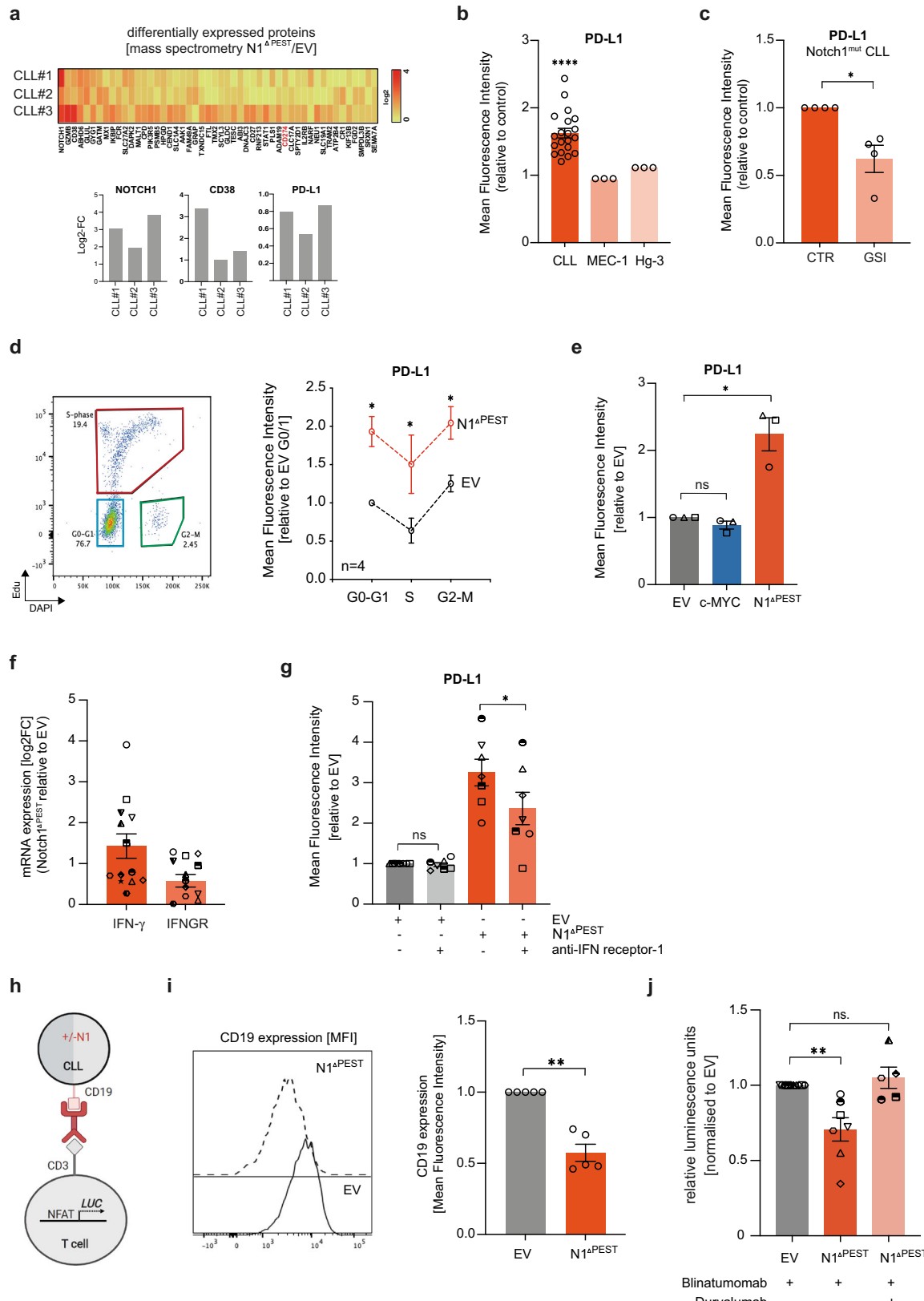

PEST-truncating mutations of *NOTCH1* were only found in 22% in the same study[19]. While the reasons for the discrepancy between the presence of NOTCH1-ICD protein and gene mutations remains elusive, these data also indicate that carrying a *NOTCH1* mutation is fundamentally different from expressing NOTCH1-ICD as only gene mutations appear to predict for a poor clinical outcome. Furthermore, these

data also hint on the importance of the tumor microenvironment for NOTCH1 activation, suggesting that mutations of oncogenes still rely on signals from non-malignant cells to fully unfold their detrimental effects. Our limited knowledge about the contributing factors for activation indicate that the lymph node environment is predominantly important for its activation[21,27]. Our experimental system to only

**Fig. 5 | NOTCH1^ΔPEST up-regulates PD-L1 on CLL cells. a** Heatmap of the top 50 differentially expressed proteins identified by mass spectrometry. Histogram with the Log2FC values from three independent patients analyzed is shown. **b** PD-L1 expression analyzed on GFP-positive cells 5 days post transduction and quantified as ratio of MFI of CLL (*n* = 20) and cell lines (*n* = 3 repeats) transduced with NOTCH1^ΔPEST or an empty vector control. **c** Quantification of PD-L1 on *NOTCH1*-mutated CLL cells treated for 48 h with a γ-secretase inhibitor (10 μM) or control (DMSO) (*n* = 4; *p* = 0.032). **d** PD-L1 expression quantified as ratio of MFI for CLL cell transduced with an *empty vector* (blue) or *NOTCH1^ΔPEST* (red) during cell cycle progression (*n* = 4). Cells were pulsed for 12 h with Edu (*p* = 0.017, 0.03 and 0.01, respectively). **e** Comparison of PD-L1 expression on CLL cells, transduced with *NOTCH1^ΔPEST* or *c-MYC* relative to the empty vector control (*n* = 3; *p* = 0.037). **f** Bar graph of the Log2FC values of *IFNG* and *IFNGR1* expression analyzed by RNAseq following NOTCH1^ΔPEST transduction (*n* = 13). **g** PD-L1 expression of CLL cells (*n* = 7)

transduced with an empty vector or NOTCH1^ΔPEST and treated with a blocking IFNG receptor antibody for 24 h; (*p* = 0.011). h. Schematic diagram of the experimental setup for data shown in panel **j. i** CD19 expression on CLL cells transduced with an *empty vector* (gray) or *NOTCH1^ΔPEST* (orange) 5 days post transduction (*n* = 5), assessed by flow cytometry. Representative flow cytometry histogram image is shown on the left (*p* = 0.021). **j** Quantification of luciferase activity (RLU) of Jurkat NFAT-Luc reporter cells co-cultured with empty vector (gray bar), NOTCH1^ΔPEST (orange bar) or NOTCH1^ΔPEST CLL cells in the presence of 10 nM Blinatumomab and 0.1 nM Durvalumab (pink bar). CLL cells (*n* = 7) were co-cultured with Jurkat cells at a ratio of 2:5 for 24 h (*p* = 0.0226). Cohorts are shown as mean ± SEM. Statistical analysis was done by multiple paired *t*-tests (**d**), paired *t*-tests (**b, c, i**) or one-way ANOVA followed by paired *t*-tests (**e, g, j**). Each symbol refers to an individual patient sample. *P* < 0.05, **P* < 0.01, ***P* < 0.001, ****P* < 0.0001 and not significant (ns) *P* > 0.05.

express PEST-deficient ICD of NOTCH bypassed microenvironment requirements for activation and thereby revealed an 'unbiased' NOTCH1/2-specific program. Notably, this approach does not account for signal modulation originating from the extracellular domains of NOTCH, which could possibly further amplify or buffer signal strength.

Our experimental system to retrovirally infect primary malignant B cells mimics a lymph node environment and therefore allows for studying genes in activated, proliferative cells. This contrasts with many CLL studies, which are commonly done with non-cycling cells derived from the peripheral blood. Although such studies provide relevant answers to understand the disease biology, they are also likely to miss important aspects relevant to disease progression, driven by proliferative cells. The regulation of PD-L1 illustrates the limitations of studies with non-cycling cells, as we observed that recently egressed cells from lymph nodes already downregulated PD-L1 expression, which becomes almost undetectable in the peripheral blood. We speculate that the magnitude of NOTCH-induced gene expression changes is dependent on cell-migration, the relative abundance of NOTCH1-target genes and their stability. Therefore, we postulate that no unique "NOTCH-gene signature" exists as there are likely multiple, which are context- and tissue-dependent and their stability will largely be determined by the absence or presence of *NOTCH* mutations. This idea is supported by our clinical data, suggesting the presence of at least two NOTCH1 gene signatures in peripheral blood cells of relapsed patients and only indicative of a poor prognosis if associated with low levels of *CIITA*.

The negative prognostic effect of *NOTCH1* mutations in CLL becomes even more evident if they occur on the background of trisomy 12. NOTCH1 mutations are enriched in patients with trisomy 12[31,32], suggesting that this chromosomal aberration provides a selective advantage for *NOTCH1* mutations. In addition, the risk for patients with trisomy 12 for transformation into clonally related Richter's syndrome is 10-times higher for *NOTCH1*-mutated patients compared to wild-type, suggesting that Notch1 signaling drives genomic instability and clonal evolution[14,55]. Our method to generate isogenic primary CLL cells provided an opportunity to directly address this question. Unexpectedly, our experiments did not identify a distinct NOTCH1-regulated gene set, present only in trisomy 12 cells, but rather indicated a higher amplitude of gene-regulation in trisomy 12 compared to del13q cells. This observation raises a further question of what other factors determine the selective advantage for clones concurrently harboring trisomy 12 and *NOTCH1*-mutations? A possible explanation is the observation that trisomy 12 CLL cells have an increased expression of CD29, CD49d and ITGB7, which occurs independently of *NOTCH1* mutations[56] and allows for an improved adherence to cells of the microenvironment. As an immediate consequence and since *NOTCH1*-mutated cells are still dependent on ligand-binding for activation, trisomy 12 cells may experience prolonged NOTCH1 signaling. Therefore, and based on our data, we propose that in the subgroup of patients with trisomy 12 the selective advantage for *NOTCH*-mutations

is based on enhanced, ligand-mediated NOTCH1 activation, rather than due to a specific genetic program governed by NOTCH1.

Gene repressive functions of NOTCH1 have previously been underappreciated. Our data indicated that Notch-signaling permits immune escape of malignant B cells through downregulation of *HLA-class II* expression. The prognostic significance of HLA-class II expression is well documented for DLBCL[57] and PMBCL[41] and shows that short overall survival is associated with low expression levels in these entities. Similar to our study, low HLA-expression levels were not due to large genetic deletions on chromosome 6 but correlated with *CIITA* expression levels[40], pointing to transcriptional de-regulation of HLA class II genes in high-grade lymphoma.

Our data suggest that *CIITA* expression levels are a prognostic marker for indolent B-cell malignancies and show that NOTCH1 is a strong epigenetic suppressor of *CIITA* transcription. NOTCH1-mediated control of *CIITA* expression has not previously been reported and the underlying mechanisms of this regulation remain to be defined. Since we observed that NOTCH1 signaling also upregulated *IFN-γ*, which itself can activate *CIITA* transcription[58], the transcriptional repression of *CIITA* by NOTCH1 likely involves epigenetic silencing of its promotors as shown for other hematological malignancies[59]

The NOTCH1-mediated modulation of surface receptors regulating the interaction with T cells expectedly has effects on the composition of the tumor microenvironment. We found significantly more cycling CLL cells in proliferative lymph nodes of NOTCH1-expressing cells compared to non-expresser, associated with an increased number of CD4+ and CD8+ T cells. This association is likely to be mediated through the recruitment of T cells through the secretion of CCL3 and CCL4, derived from activated CLL cells[60,61]. Notably, an increased number of CD8+ T cells, but not of CD4+ T cells, was associated with higher expression of PD-1 in NOTCH1 positive samples, indicating terminal differentiation and exhaustion of CD8+ cells. The role of CD4+ T-cell subsets in CLL is far from being completely understood, but the collective evidence indicates that CD4+ T cells overall are tumor-promoting. This conclusion is based on the dependency of CLL cells on autologous T cells to engraft in NSG mice[62], in vitro growth-promoting effects of CLL-specific Th1-cells[63] and correlation between higher CD4+ cell counts and shorter PFS and OS[64]. The relative contribution of individual CD4+ subsets is less clear, but numerous studies suggest that this phenotype may predominantly be driven by Tregs[65,66], possibly through their secretion of pro-leukemic cytokines such as TGFβ and IL-10. While the direct comparison of data from human to mouse always requires caution, data from our experiments in NSG mice indicate that Notch1 signaling in CLL cells drive the expansion of CD4+ T cells, in keeping with those studies.

In contrast to the CIITA-HLA-class II axis, the role of PD-L1 for the suppression of T cell functions in CLL is better defined. Pre-clinical data indicate that the PD-1/ PD-L1 axis actively contributes to immune escape, demonstrating that PD-L1 inhibition prevented the development of a CLL-like disease in the Eμ-TCL1 mouse model[67]. This

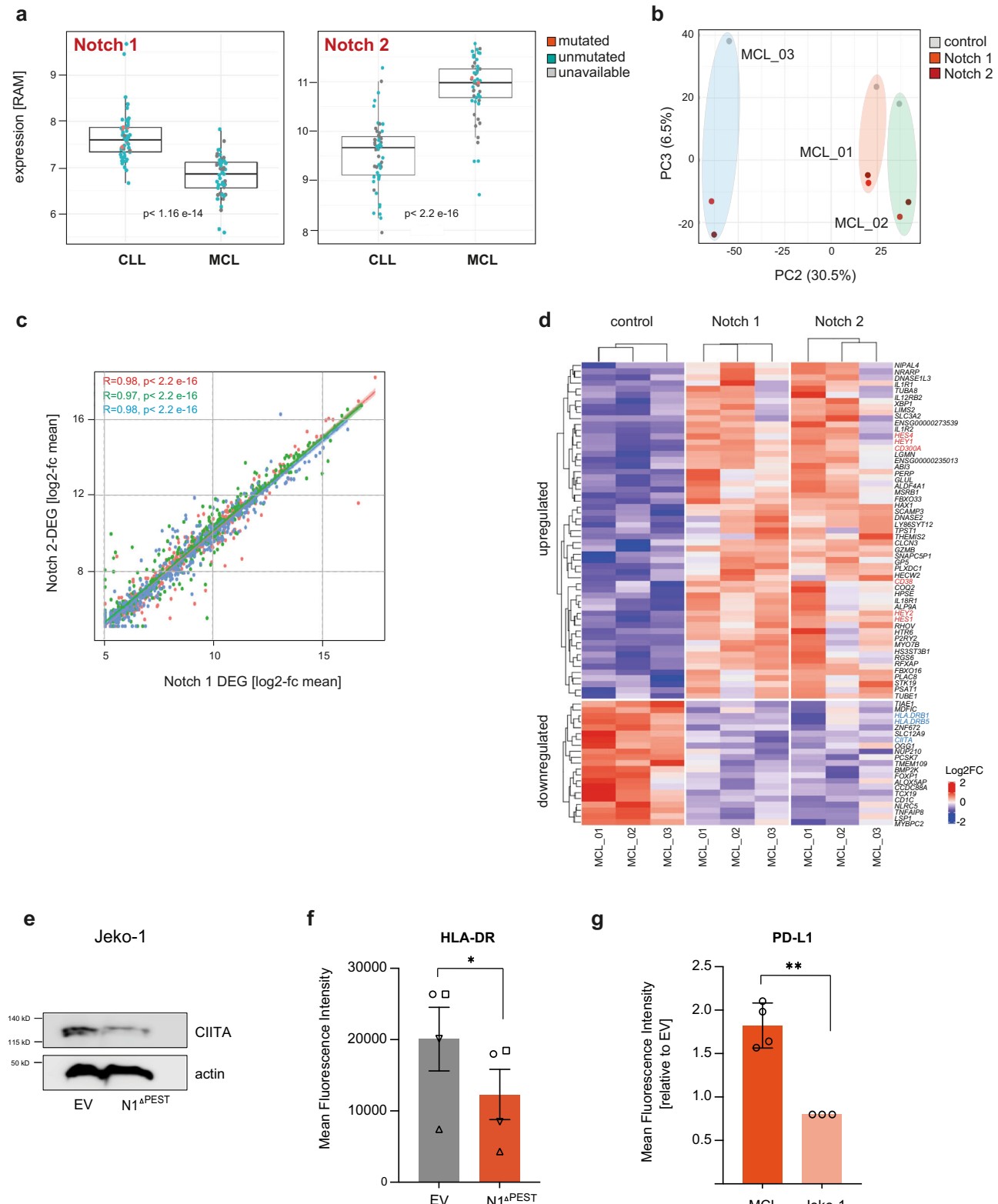

therapeutic effect was further enhanced through a simultaneously administered BTK-inhibitor[68]. Additionally, PD-1 blockade restored normal immune synapse formation between T and CLL cells[69]. A recent study demonstrated that activation of autologous T cells with an E3-ligase inhibitor also reverted PD-L1 mediated suppression of cytotoxic T cells, causing anti-tumor effects in a CLL xenograft model[70]. Although these pre-clinical data strongly suggest that immune checkpoint blockade is therapeutically useful, clinical data supporting

this have been inconclusive to date. A phase II clinical trial with pembrolizumab, a humanized anti-PD-1 antibody, failed to show objective responses in non-transformed CLL. However, the same study reported overall response in 44% of patients with Richter's transformation (RT)[71]. The reasons for this discrepancy are unknown; enhanced immunogenicity of RT cells may be based on the presentation of tumor-neoantigens, generated in the process of transformation. While the presence of *NOTCH1* mutations is strongly associated with

**Fig. 6 | NOTCH1 and NOTCH-2 ICD regulate similar transcriptomes in MCL.**
**a** Box plots (1st to 3rd quartile, center = median) showing the expression levels of *NOTCH1* and *NOTCH2* genes in a series of CLL ($n = 54$) and MCL ($n = 53$) primary samples. The *NOTCH1* or *NOTCH2* mutational status of each sample is color labeled as indicated. RAM = robust multi-array average. **b** Unsupervised principal component analysis (PC2/3) of the RNAseq profiles of $n = 3$ paired MCL primary cases transduced with either an empty vector control, a NOTCH1$^{\Delta PEST}$ or a NOTCH2$^{\Delta PEST}$ expressing vector. A matrix of 40,490 transcripts was used to generate the unsupervised PCA. **c** Scatter plot showing the gene expression of the top 500 most variable genes in NOTCH1$^{\Delta PEST}$ or NOTCH2$^{\Delta PEST}$ expressing samples ($n = 3$). Pearson correlation of their expression in the NOTCH1$^{\Delta PEST}$ and NOTCH2$^{\Delta PEST}$ samples per pair was calculated. Each color corresponds to one of the three paired independent samples analyzed. Error bands indicate the 95% percentile of the mean. **d** Heatmap showing expression levels of genes differentially expressed in control vs. NOTCH expressing MCL samples (i.e., both NOTCH1$^{\Delta PEST}$ and NOTCH2$^{\Delta PEST}$ together). Gene expression levels are indicated as row z-scores. **e** CIITA immunoblot of Jeko-1 cells 5 days post transduction with NOTCH1$^{\Delta PEST}$ or an empty vector control. Proteins were extracted following cell sorting of GFP-positive cells. One of three independent experiments is shown. **f** HLA-DR expression quantified by flow cytometry of NOTCH1$^{\Delta PEST}$ transduced MCL cells ($n = 4$). HLA-DR expression was analyzed on GFP-positive cells 5 days post transduction. Each symbol refers to an individual patient sample. ($p = 0.021$). **g** PD-L1 expression quantified by flow cytometry of transduced primary MCL ($n = 4$) and Jeko-1 cells. PD-L1 expression was analyzed on GFP-positive cells 5 days post transduction ($p = 0.003$). Cohorts are shown as mean ± SEM. Statistical analysis was done by paired *t*-tests (**f**) and unpaired *t*-test (**g**). *$P < 0.05$, **$P < 0.01$, ***$P < 0.001$, ****$P < 0.0001$ and not significant (ns) $P > 0.05$.

Richter's transformation, it remains unclear from this trial whether responding patients were carrying *NOTCH1* mutations. Our data predict that PD-1/ PD-L1 inhibition could be more efficacious in *NOTCH1* mutated patients and future prospective studies are needed to address this.

## Methods

### Primary cells and cell culture
After patients' informed consent and in accordance with the Helsinki Declaration, peripheral blood was obtained from patients with a diagnosis of CLL or MCL. Studies were approved by the Cambridgeshire Research Ethics Committee (07/MRE05/44).

PBMCs were isolated from heparinized blood samples from patients by centrifugation over a Ficoll-Hypaque layer (PAN-Biotech, Aidenbach, Germany). Purity of CLL population was assessed by flow cytometry and only samples with >85% CD19$^+$CD5$^+$ were used. After harvesting, malignant B cells were either frozen down as viable cells or directly cultured in Advanced Roswell Park Memorial Institute medium (Advanced RPMI-1640; Invitrogen, Carlsbad, CA) with GlutaMAX containing 10% FBS (Gibco), 100 IU/ml penicillin and 100 µg/ml streptomycin and kept at 37 °C in a humidified incubator (5% CO$_2$ and 95% atmosphere). We have not observed differences in transduction efficacy between fresh and frozen cells.

Autologous patient-derived T cells were isolated using CD3 MicroBeads (Miltenyi Biotec) according to manufacturer instructions. Purified cells were then cultured for 7 days at the density of 10$^6$ cells in RPMI supplemented with 10% FBS, 1000 unit/mL IL-2 (R&D systems), anti-CD3 (2 µg/ml), anti-CD28 (4 µg/mL) 100 IU/ml penicillin and 100 µg/ml streptomycin and kept at 37 °C in a humidified incubator.

MM1 feeder cells were cultured in MEM Alpha+GlutaMAX medium (ThermoFisher Scientific, Winsford, UK) supplemented with 10% FBS (Gibco), 10% horse serum (Sigma-Aldrich, Dorset, UK), 10 µM 2-ME and 1% penicillin/streptomycin (Gibco). For co-culture experiments, feeder cell were seeded in a 12 multi-well plate coated with 0.1% Gelatin 24 h before the addition of primary CLL cells at a concentration of 10$^6$ cells/ml.

Cell lines MEC-1, Hg-m3, Jeko-1 and Jurkat were cultured in RPMI-1640 (Invitrogen, Carlsbad, CA) supplemented with 10% fetal calf serum (Gibco), 100 IU/ml penicillin and 100 µg/ml streptomycin (Gibco). Lenti-x-293 Cell Line (Clontech Laboratories, 632180) were cultured in Dulbecco's modified Eagle's medium (DMEM, Invitrogen, Carlsbad, CA) containing 10% FBS, 100 IU/ml penicillin and 100 µg/ml streptomycin and kept at 37 °C in a humidified incubator (5% CO$_2$ and 95% atmosphere). All cell lines used in this study were tested to be free from mycoplasma.

### Mouse model
8–10-week-old male NOD.*Cg-Prkdc$^{scid}$Il2rg$^{tm1Wjl}$*/SzJ (NSG) mice injected intraperitoneally with 10$^7$ retrovirally transduced CLL cells and 5*10$^5$ autologous T cells (20:1 CLL:T cell). Following close monitoring for 3 weeks mice were culled and spleen and peritoneal cavity fluid were harvested for the analysis of human cell engraftment.

These animal studies have been regulated under the Animals (Scientific Procedures) Act 1986 Amendment Regulations 2012 following ethical review by the University of Cambridge Animal Welfare and Ethical Review Body (AWERB-PPL number P846C00DB).

### Flow cytometry
Cells were stained with fluorophore-labeled antibodies in 2% BSA in PBS according to the manufacturer's instructions. For apoptosis analysis, conjugated Annexin-V and DAPI were used for the detection of apoptotic cells according to the manufacturer's instructions. Cell cycle analysis was performed using the Click-iT™ EdU Alexa Fluor™ 647 Flow Cytometry Assay Kit (ThermoFisher Scientific) according to the manufacturer's instructions. Cells were pulsed with 10 µM Edu for 12 h.

Calcium-Flux assay was performed using $5 \times 10^6$ cells. Fluo-4 (5 µM) (ThermoFisher Scientific) was added to 500ul of cells in serum-free media and incubated for 15 min at room temperature with protection from light. Cells were then washed and re-suspended in 100 µl HBSS (Ca2+ free) plus 20 µg biotin-SP Affini-Pure Fab Fragment Goat Anti-Human IgM for 20 min on ice. Cells were then washed, re-suspended in 500 µl HBSS and incubated for 20 min at 37 °C. DAPI was added to identify dead cells. Samples were analyzed on flow cytometry. Initial measurement was lasting for 20 seconds to record baseline Ca$^{2+}$ signal, then 20 µl streptavidin (1 mg/ml) was added to stimulate the Ca$^{2+}$ flow. Measurement was resumed for up to 180 s.

Samples were acquired on a LSRFortessa™ X-20 cell analyzer (BD Biosciences, Oxford, UK) and analyzed using FlowJo software version 10 (Tree Star).

FACS cell sorting was performed using the BD Influx™ Cell Sorter (BD Biosciences). Gating strategy is presented in Supplementary Fig. 8.

### Retroviral transduction
Feeder MM1 cells were engineered to stably express human IL-21, human CD40L, and human BAFF and plated in 12 multi-well plates at $5 \times 10^4$/ml. After 24 h CLL cells were defrosted and co-cultured on MM1 cells for 72 h at $1 \times 10^6$/ml. In order to produce retroviral supernatant packaging plasmids and envelopes were used as follows. -Gag-pol: 1.5 µg, -Envelope: 1 µg GaLV, 1 µg FeLV or 1 µg SsAV, - Retroviral cDNA construct: 4 µg mixed in 1 ml of Opti-MEM media (Invitrogen) and 22 µl of TransIT-293 (Mirus) was used to transfect Lenti-x-293 (10 cm$^2$ dish). After 48 h, the cell supernatant was filtered through a 0.45 µM filter and the viral supernatant was added to the primary cells/ feeder cells co-culture. Transduction was performed by spinoculation centrifugation (1500 × *g*, 2 h at 32 °C) with the addition of 10 µg/ml Polybrene (INSIGHT Biotechnology) in 12 well plates. Viral supernatant was replaced with fresh media 4 h after centrifugation for retroviral infection. Cells were then maintained at 37 °C with 5% CO$_2$ for at least 3 days before FACS analysis.

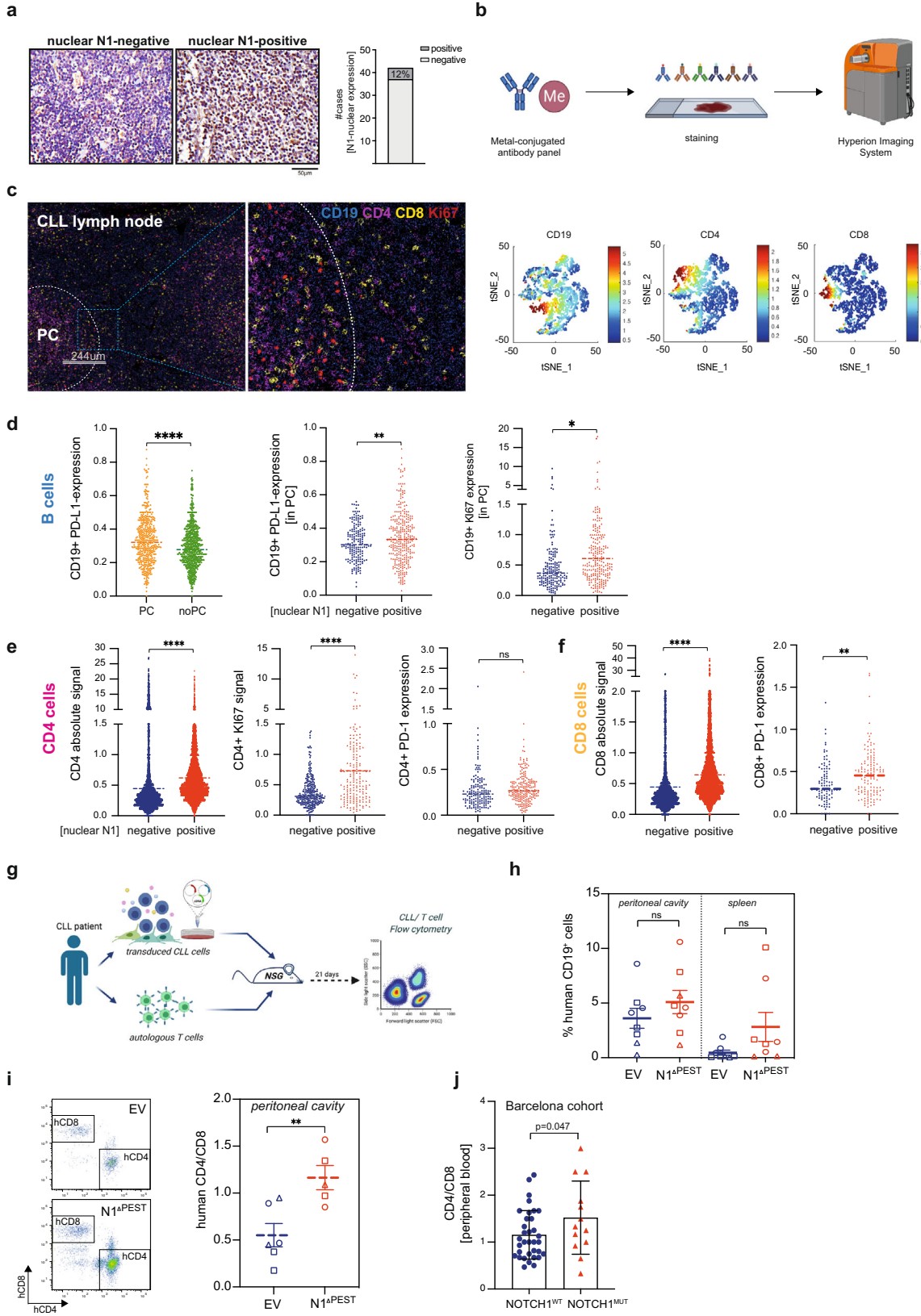

## Plasmids

For overexpression of human genes, CDS sequences were cloned into the MSCV-IRES-GFP, pMSCV-IRES-GFP II (Addgene #52107) and MSCV-IRES-Thy1.1 DEST (Addgene #17442) vectors. The truncated NOTCH1^ΔPest vector was created by adding a stop codon at the start of the PEST domain to the full ICN1 sequence.

To generate fusion envelope constructs, pHIT123 (obtained from D.H) containing the retroviral ecotropic envelope, human cytomegalovirus immediate-early promoter and the origin of replication from simian virus 40 was used as the backbone. The viral envelope sequences for GaLV, FeLV (GenBank: K01209.1), SSaV (GenBank: AF055064.1) were purchased from Integrated DNA

**Fig. 7 | NOTCH1 activation is associated with T-cell proliferation in vivo.**
**a** Representative IHC staining of NOTCH1 in CLL lymph node biopsies. **b** Graphical scheme of the mass cytometry analysis from nuclear NOTCH1 negative ($n = 3$) or positive ($n = 4$) specimens. Created with BioRender.com. **c** Multiplexed IMC image example of a NOTCH1 positive specimen with an enlarged area of the proliferative center (PC). Representative t-SNE plots from an individual patient specimens are presented for CD19, CD4, and CD8 cell populations ($n = 4$). **d** Intensity of cellular signal per given cell was calculated using the HistoCat software. PD-L1 signal in CD19-gated cells in PC and non-PC areas ($p < 0.0001$), PD-L1 ($p = 0.0079$) and KI67 ($p = 0.011$) in CD19-gated PC-cells from samples with a positive or negative NOTCH1 nuclear staining. **e** Total CD4 signal, KI67, and PD-1 in CD4-gated cells in PCs of NOTCH1 positive or negative samples. **f** Total CD8 signal and PD-1 in CD8-gated cells in PCs of NOTCH1 positive or negative samples (**$p = 0.0044$). **g** Graphical scheme of the in vivo experiment. CLL cells were transduced with an *empty vector*

or *NOTCH1$^{\Delta PEST}$* and intraperitoneally injected into male 8–10 week-old NSG mice. Autologous T cells were cultured with IL-2, α-CD3, and α-CD28 for 7 days prior to injection. Created with BioRender.com. **h** Engraftment of human CD19$^+$ cells in the peritoneal cavity (left) and spleen (right) of mice injected with NOTCH1$^{\Delta PEST}$ ($n = 8$) or an empty vector ($n = 8$) transduced CLL cells. In total 16 mice were analyzed using cells from 3 independent donors. Each symbol refers to an individual patient sample. **i** Quantification of human autologous CD4$^+$/CD8$^+$ T cells. Mean value was obtained from 3 independent experiment using different donor cells. Each symbol refers to an individual patient sample ($p = 0.007$). **j** Ratio of human CD4$^+$/CD8$^+$ T cells in the peripheral blood of a cohort of treatment-naive CLL patients with mutated ($n = 13$) or wild-type *NOTCH1* ($n = 34$). Cohorts are shown as median (**d**–**f**) or mean ± SEM (**h**–**j**). Statistical analysis was done by paired (**h**, **i**) or unpaired *t*-tests (**d**–**f**, **j**) *$P < 0.05$, **$P < 0.01$, ***$P < 0.001$, ****$P < 0.0001$ and not significant (ns) $P > 0.05$.

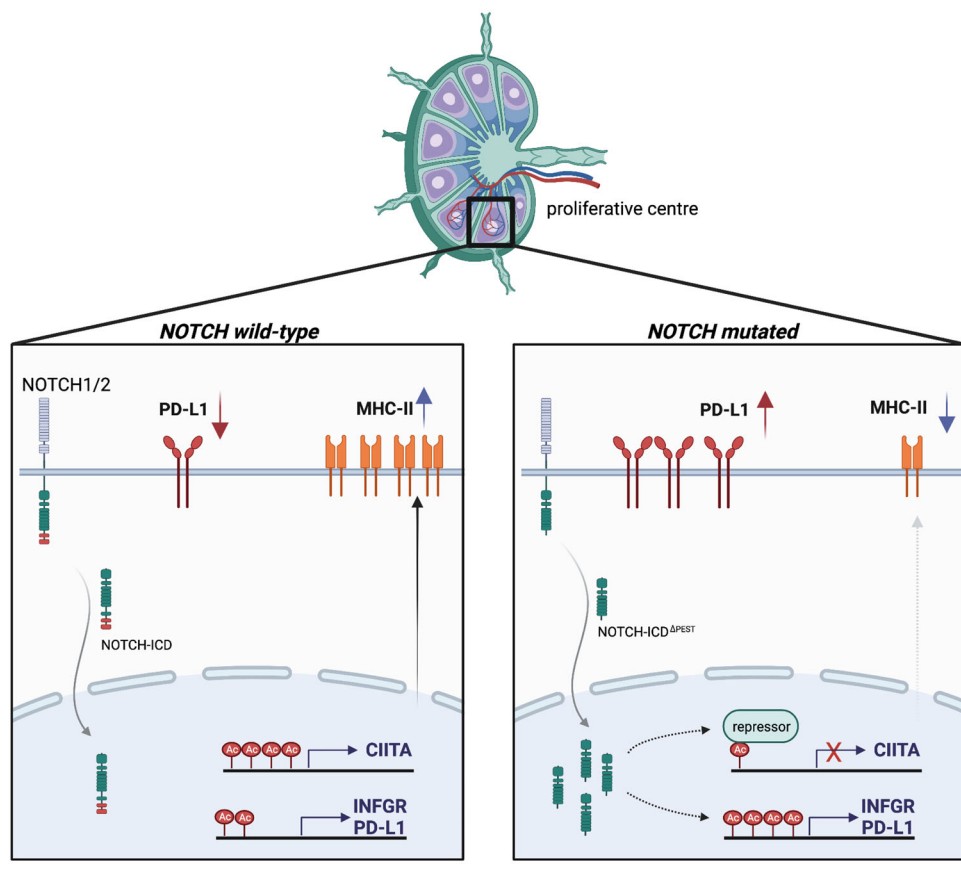

**Fig. 8 | Graphical summary.** Left panel: In the absence of *NOTCH* mutations, tumor B cells in proliferative centers constitutively express MHC class II and PD-L1 on their cell surface. Right panel: Mutations in the PEST domain of *NOTCH1* or *NOTCH2* cause higher NOTCH levels and transcriptional activity, which leads to a significant downregulation of MHC class II proteins through epigenetic suppression of *CIITA*. PD-L1 levels increase, partly due to enhanced transcription of the interferon-γ-receptor and an autocrine feedback loop involving INFγ. Created with BioRender.com.

Technologies (IDT, Iowa, USA) as synthetic double-stranded DNA and inserted by Gibson assembly. All plasmids were verified by sanger sequencing.

## T-cell reporter assay
T-Cell Activation Bioassay (Promega) was performed according to the manufacturer's instructions. Four days post transduction CLL cells were sorted for GFP$^+$ and CD19$^+$ expression. Following sorting, $2 \times 10^4$ CLL cells where then co-cultured with $5 \times 10^4$ Jurkat NFAT reporter cells with the addition of Blinatumomab (10 nM) and 0.1 nM Durvalumab (Stratech Scientific) in white walled 96 wells plate (Corning). TCR-mediated luminescence was measured 24 h later using SpectraMax M5e Microplate Reader.

## Expression analysis/qPCR
Total RNA was isolated using the RNeasy Mini Kit (Qiagen, Manchester, UK), and complementary DNA (cDNA) was obtained using the qScriptTM cDNA SuperMix kit (QuantaBio, Beverly, MA, USA). Quantitative reverse-transcription polymerase chain reaction (RT-qPCR) was performed on isolated mRNA using the fast SYBR reagents and the Applied Biosystems™ QuantStudio™ 12 K system. Target gene expression levels were normalized to GAPDH and values are represented as fold change relative to control using the ΔΔCt method. Primers used are listed in Supplementary Table 1.

## Western blot
Cultured cells were collected and lysed with RIPA buffer and a total of 20 μg protein was separated by SDS-polyacrylamide gel

electrophoresis using 4–12% NuPAGE Bis-Tris gels (ThermoFisher Scientific), blotted to polyvinylidene difluoride (PVDF) membranes (Millipore), and probed with primary antibodies (B-Actin-HRP (Cell Signaling), CIITA 7-1H (Insight Biotech)). Images were captured with the Azure Biosystem c300 (Dublin, CA, USA) digital imaging system.

## Mass spectrometry

Following cell sorting $10^6$ cell pellets were collected for mass spectrometry analysis. Protein isolation and TMT-6plex labeling was performed as described previously[72]. In brief, TMT mix was fractionated on a Dionex Ultimate 3000 system at high pH using the X-Bridge C18 column (3.5 μm, 2.1 × 150 mm, Waters) with 90 min linear gradient from 5% to 95% acetonitrile containing 20 mM ammonium hydroxide at a flow rate of 0.2 ml/min. Peptides fractions were collected between 20–55 min and were dried with speed vac concentrator. Each fraction was reconstituted in 0.1% formic acid for liquid chromatography-tandem mass spectrometry (LC–MS/MS) analysis.

Peptide fractions were analyzed on a Dionex Ultimate 3000 system coupled with the nano-ESI source Fusion Lumos Orbitrap Mass Spectrometer (Thermo Scientific). Peptides were trapped on a 100 μm ID × 2 cm microcapillary C18 column (5 μm, 100 A) followed by 2 h elution using 75 μm ID × 25 cm C18 RP column (3 μm, 100 A) at 300 nl/min flow rate. In each data collection cycle, one full MS scan (380–1500 m/z) was acquired in the Orbitrap (120 K resolution, automatic gain control (AGC) setting of $3 × 10^5$ and Maximum Injection Time (MIT) of 100 ms). The subsequent MS2 was conducted with a top speed approach using a 3-s duration. The most abundant ions were selected for fragmentation by collision induced dissociation (CID). CID was performed with a collision energy of 35%, an AGC setting of $1 × 10^4$, an isolation window of 0.7 Da, a MIT of 50 ms. Previously analyzed precursor ions were dynamically excluded for 45 s. During the MS3 analyses for TMT quantification, precursor ion selection was based on the previous MS2 scan and isolated using a 2.0 Da m/z window. MS2–MS3 was conducted using sequential precursor selection (SPS) methodology with the top10 settings. For MS3, HCD was used and performed using 65% collision energy and reporter ions were detected using the Orbitrap (50 K resolution, an AGC setting of $1 × 10^5$ and MIT of 105 ms).

Peptide intensities were then normalized using median scaling and protein level quantification was obtained by the summation of the normalized peptide intensities. A statistical analysis of differentially regulated proteins was carried out using the limma R-package from Bioconductor[73] Multiple testing correction of p-values was applied using the Benjamini-Hochberg method (https://www.jstor.org/stable/2346101?seq=1#page_scan_tab_contents) to control the false discovery rate (FDR). The mass spectrometry proteomics data have been deposited to the ProteomeXchange Consortium via the PRIDE[74] partner repository with the dataset identifier PXD024112.

## Mass cytometry

FPPE lymph node Sections 5 μm in thickness were cut with a Leica CM 1850 UV cryomicrotome and processed according to the manufacturer's instruction. In brief, slides were baked for 2 h at 60 °C before dewaxing with xylene and hydration with descending grades of ethanol. Tissue sections were then incubated with the antigen retrieval solution (pH 9) (Abcam) for 30 min, blocked with 3% BSA in PBS for 45 min at room temperature and then incubated with Fluidigm pathologist-verified Maxpar antibodies overnight at 4 °C in a humidified chamber. The following day, the slides were washed in 0.2% Triton X-100, followed by PBS and then stained with DNA intercalator-Ir (1:2000 dilution; Fluidigm) for 30 min at room temperature. Slides were washed in distilled deionized water and air-dried for ~30 min. Slides were inserted into the Hyperion Imaging System (Fluidigm) for data acquisition. (https://www.fluidigm.com/binaries/content/documents/fluidigm/resources/imaging-mass-cytometry-staining-for-

ffpe-sections-400322-pr/imaging-mass-cytometry-staining-for-ffpe-sections-400322-pr/fluidigm%3Afile).

For the staining we used the following Fluidigm pathologist-verified Maxpar antibodies: Anti-CD19 (6OMP31)-142$^{Nd}$; -Anti-Human CD4 (EPR6855)-156$^{Gd}$; -Anti-Human CD8a (D8A8Y)-162$^{Dy}$; -Anti-Human PD-1 (EPR4877(2))-165$^{Ho}$; -Anti-Ki-67 (B56)-168$^{Er}$; -Anti-Human PD-L1 (E1L3N)-150$^{Nd}$; -Anti-Pan-Actin (D18C11)-175$^{Lu}$; -Anti-Histone 3 (D1H2)-176$^{Yb}$

Images acquired with the Hyperion Imaging System were reviewed and single ROI were exported using MCD Viewer (Fluidigm v1.0.560.6). Single cell segmentation was performed using the open-source software CellProfiler v4.2.4 (Broad Institute). For this, individual nuclei were identified using the DNA staining intercalator-Ir and Histone H3 marker followed by identification of the cellular region by a circle of a defined radius. From this, we could now measure the intensity in each channel, and thus a proxy of the expression level of the protein in each individual cell. As the dynamic ranges of the different channels vary considerably our analysis was limited to the comparison of each single channel across the different tissue sections. In order to obtain the intensity of each channel we used the Histology Topography Cytometry Analysis Toolbox v1.761 (HistoCat) software[51]. Area with high density of KI67 expression were considered proliferation centers. t-SNE analysis across the markers of interest was created and single channel heatmaps were generated in order to gate on specific cell types. Raw data of each population was extracted into excel files and plotted using GraphPad Prism 9.0 (GraphPad Software, La Jolla, USA).

## RNAseq

Total RNA was isolated from GFP$^+$CD19$^+$ sorted cells using the RNeasy Mini Kit (Qiagen, Manchester, UK). Samples (25 ng total RNA) were then processed for NGS sequencing using the NuGEN TRIO Kit (NuGEN) and the size distribution of the resulting libraries was analyzed on Agilent Bioanalyzer HS DNA chips. A single library pool containing all samples was generated for sequencing and quantified using the NEB Library Quant kit, a SYBRgreen-based qPCR method.

The sequencing of the CLL library pool was performed in two runs on both lanes of a HiSeq 2500 RapidRun flow cell in the paired-end mode: 101 cycles for read 1, 9 cycles for the index read and another 101 cycles for read 2. Both runs generated excellent read qualities and quantities as indicated by the Illumina SAV software tool. Bcl-to-Fastq conversion and de-multiplexing of the reads were performed with the Illumina CASAVA 1.8.2 software using standard settings.

For all analyses, quality checks were performed using FastQC (https://www.bioinformatics.babraham.ac.uk/projects/fastqc). The alignment to the reference genome (human genome hg38, genome assembly GRCh38.p13) was done using STAR version 2.5.2a (https://doi.org/10.1093/bioinformatics/bts635). Pre- and post-alignment quality checks were summarized using MultiQC. Gene expression counts were obtained using featureCounts version v1.6.0 (https://doi.org/10.1093/bioinformatics/btt656). Additional quality checks include MA plots and heatmaps representing the Jaccard Similarity Index (JSI).

The normalization of expression levels was performed using quantile normalization using the function normalize.quantiles from the R package preprocessCore (https://github.com/bmbolstad/preprocessCore), followed by edgeR internal normalization. The differential expression analysis was performed using the standard functions from edgeR pipeline, version 3.28.0.

The Fold Change (FC) of the normalized counts of all the genes per pair of samples was calculated as B/A, where B are the NOTCH1 mutated samples and A the control samples. The differentially expressed (DE) genes were identified as follow: upregulated those genes that had a log2FC value > 0.5 in at least half of the samples while as downregulated the ones with a log2FC value < −0.5 in at least half of the samples. A more stringent cut-off was used for groups of samples

with tri12 or del13q mutation as indicated in the manuscript. As upregulated genes were characterized those genes that had a log2FC value > 0.5 in at least half of the samples and >0 in all the samples, while as downregulated the ones with a log2FC value < −0.5 in at least half of the samples and <0 in all the samples. The Pearson correlation coefficients were calculated between the expression of the DE genes of the two groups.

The sequencing of the MCL library pool was performed using Novaseq. Quality control of the fastq files was performed using fastQC. The rRNA reads were then filtered with Sortmerna 4.3.2 using the default rRNA databases. The resulting non-rRNA reads were trimmed using trimmomatic −0.39. The final reads were quantified using Kallisto quant function with default parameters. The resulting estimated counts were VST normalized and used in further downstream analyses. Generation of Principal Component Analysis was performed using VST normalized expression values with all the genes (n = 40.490). Differential expression analysis of the empty vector control vs NOTCH1$^{\Delta PEST}$/NOTCH2$^{\Delta PEST}$ samples was performed using DESeq2 with the raw expression values as input. A minimal filtering of 10 counts across all samples was applied prior to the analysis, resulting in 20.707 genes. To generate the correlation among the different conditions, the standard deviation across all samples was calculated. The genes were subsequently ordered by their SD values and the highest 500 were isolated. The Pearson correlation of the expression of these genes in the NOTCH1$^{\Delta PEST}$ and NOTCH2$^{\Delta PEST}$ samples per pair was calculated.

### H3K27ac ChIP-seq
$10^6$ cells were FACS sorted 5 days following transduction. The samples were immediately cross-linked in 1% Formaldehyde and the reaction was then quenched with 0.125 M Glycine. ChIP-seq for H3K27ac were generated following the Blueprint protocol (https://www.blueprint-epigenome.eu/index.cfm?p=7BF8A4B6-F4FE-861A-2AD57A08D63D0B58) using the antibody: C15410196/pAb-196-050 (Diagenode). The fastq files of the ChIP-seq data were aligned to genome build GRCh38 (using bwa 0.7.7, picard and samtools) and wiggle plots were generated (using PhantomPeakQualTools) according to the Blueprint pipeline (http://dcc.blueprint-epigenome.eu/#/md/methods).

Peaks of the H3K27ac data were called as described (http://dcc.blueprint-epigenome.eu/#/md/methods) using MACS2 (version 2.0.10.20131216). For all samples, H3K27ac peaks were called without input control. A set of consensus peaks for all the samples was generated by merging the locations of the separate peaks per sample. Variance Stabilized Transformed (VST) values were calculated for the consensus peaks using DESeq2. For downstream analyses, only peaks present in at least 2 samples were used (45.300 peaks). All Principal Component Analyses (PCAs) were generated with the prcomp function using corrected VST values.

For the isolation of the peaks forming principal component 5 (PC5), the Pearson correlation coefficients between the eigenvalues of PC5 and each peak across all samples were calculated. Correlation coefficients with a p value <0.05 were included, resulting in 587 peaks. Using previously reported chromatin states of reference CLL samples, peaks located in inactive chromatin regions were removed resulting in 484 peaks. All known genes of the hg38 annotation were downloaded using the GenomicFeatures package, and their locations were extended by 1.5 kb upstream to include their promoter region. The peaks were subsequently annotated according to overlaps with the genes' coordinates, resulting in 422 peaks. The rtracklayer package was used for the import of the H3K27ac signal files of the samples. The subtracted signal was calculated in each pair of samples (NOTCH1 mutated - control) per 1 bp. The bedGraphToBigWig application was used for the transformation of those regions to signal files appropriate for loading to the UCSC browser.

### Patients validation cohort
Data from a published gene expression dataset[34] was re-analyzed based on NOTCH1 mutational status and genetic background. Differentially expressed genes between NOTCH1 mutated samples and NOTCH1 wild-type samples were identified using DESeq2 (https://genomebiology.biomedcentral.com/articles/10.1186/s13059-014-0550-8). The R package FGSEA (http://bioconductor.org/packages/release/bioc/html/fgsea.html) was then used for gene set enrichment analysis against gene defined as upregulated and downregulated after NOTCH1 overexpression by RNAseq.

### Clinical association of CIITA expression
CIITA expression was correlated with genes indicating NOTCH-pathway activation (averaged expression levels of *HES1/2*, *HEY1/2*) on n = 337 treatment-naive patients. NOTCH-pathway activation and *CIITA* expression levels were inversely correlated.

Clinical impact for cases with high NOTCH-pathway activation (averaged expression levels of *HES1/2*, *HEY1/2*, expression above median expression level was defined as high NOTCH-pathway activation) and corresponding high or low *CIITA* expression levels (median high vs. median low *CIITA*) was assessed using gene expression data generated from PBMCs in a cohort of fludarabine-resistant CLL patients.

### Statistical analyses
Data analyses were performed using GraphPad Prism 9.0 (GraphPad Software, La Jolla, USA) with unpaired or paired analyses as indicated. For experiments where more than two groups are compared, statistical analyses were performed using one-way ANOVA followed by two-tail Student t-tests. Statistical annotations were denoted with asterisks as follows: ****$P < 0.0001$, ***$P < 0.001$, **$P < 0.01$, *$P < 0.05$, and not significant (ns) $P > 0.05$.

### Reporting summary
Further information on research design is available in the Nature Research Reporting Summary linked to this article.

## Data availability
The sequencing and proteomic data generated in this study are available under the following links: Raw RNAseq data: GEO GSE150610. CHIP-seq data: European Genome-Phenome Archive: EGAS00001005793: https://eu-central-1.protection.sophos.com?d=ega-archive.org&u=aHR0cHM6Ly9lZ2EtYXJjaGl2ZS5vcmcvc3R1ZGllcy9FR0FTMDAwMDEwMDU3OTM=&i=NjJjZTQ0N2ZiNjkxN2ExMDJlZTQ0NWFm&t=akZqYmljngwWDByYmhpemxkazBGNGFIdXVJV0lrL2ZsVks3MDZkNWZTaz0=&h=337fca0086cf49d89eda69541a0c18cd. Raw Mass spectrometry data: ProteomeXchange Consortium (accession number: PXD024112). All other data are available within the article, its supplementary data (online) or the source data file. Source data are provided with this paper.

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

## Acknowledgements

We would like to express our deepest gratitude to patients who donated blood for research purposes. In particular, we thank Dr. Joanna Baxter and her team for enrolling patients into these studies. We also wish to thank the Cambridge NIHR BRC Cell Phenotyping Hub for their advice and support in cell sorting, Dr. Richard Grenfell (CRUK Cambridge Institute) for his support with the Hyperion tissue imager and Dr. Leigh-Anne McDuffus for her help with IMC processing and analysis. This research was funded in part by the Wellcome Trust [203151/Z/16/Z], the UKRI Medical Research Council [MC_PC_17230] and the NIHR Cambridge Biomedical Research Centre (BRC-1215-20014*). For the purpose of open access, the corresponding author has applied a CC BY public copyright licence to any Author Accepted Manuscript version arising from this submission. This work was also funded by Cancer Research UK (CRUK; C49940/A17480-I.R. was a senior CRUK fellow), Kay Kendall Leukaemia Fund (M.M-KKL1258), and Fundació La Marató de TV3 (201924-30). J.B. and S.S. are funded by the Deutsche Foschungsgemeinschaft (DFG), SFB1074 subproject B1 and B2. A.M.D. is supported by the Beatriu de Pinós Programme of the Government of Catalonia (2018-BP-00231). This work was partially developed at the Centro Esther Koplowitz (CEK, Barcelona, Spain).

## Author contributions

M.M. performed and analyzed experiments. A.M.D., S.C., and J.I.M.S. ran and analyzed the H3K27ac ChIP-seq., E.G.H. and A.R. performed and analyzed the IHC from CLL lymph nodes. G.G. performed experiments with MCL cells, J.B., J.L., S.S., and T.Z. analyzed gene expression in primary CLL samples. A.Mo. helped to perform the PDX experiment. V.N.R.F., C.S.R.C., and C.DS. ran and analyzed mass spectrometry experiments. I.M and S.C. analyzed RNAseq data. S.D. provided human CLL samples. D.H. provided models and data interpretation. This project was designed by J.I.M.S. and I.R., I.R. wrote the manuscript.

## Competing interests

The authors declare no competing interests.
