## [Peer Review File · Nature Communications]

Viral transduction of primary human lymphoma B cells reveals mechanisms of NOTCH-mediated immune escapeREVIEWER COMMENTS

Reviewer #1 (Remarks to the Author):

NCOMMS-22-01742-T

In this manuscript by Mangolini and colleagues addressed the long-standing question on how NOTCH 1/2 mutations contribute to disease progression and high-grade transformation. To perform these investigations the authors developed a new method that allows the generation of isogenic primary human cells from CLL and MCL patients. Notably isogenic primary cancer cells carrying either NOTCH1-ICD or NOTCH2-ICD up-regulate PDL-1 and lose HLA-class II suggesting that NOTCH mutations underlie immune escape which could aid high-grade transformation. These important observations were not found in cell lines, further supporting the strength of the study and of the new methodology. This reviewer has mainly comments that primarily relate to data presentation.

Fig1: It is advisable that the authors show consistency between the flow plot in (C) with the cumulative data presented.

In the cumulative data presented in (C) is each dot representing a different primary sample i.e., biological replicates or are these technical replicates? Was the same primary sample transduced with the various viral envelopes? If yes, it would be important to show the variation for each to evaluate the real difference between the various envelopes.

(E) is the data on p21 expression performed on sorted GFP cells? This is not explicit in the figure legend.

(G) The data must be presented as paired analysis between the same CLL sample comparing EV and C-MYC otherwise it is not possible to evaluate the effect.

The authors may wish to describe in more detail the co-culture system. Once cells are sorted after infection the cells are no longer added to co-culture with feeder cells, correct? E.g., in (C) when it is written in the legend "6 days post transduction" does this correspond to day 13 in the schema in (B)?

Fig2: Could the authors confirm that the statistics presented in (B) are correct? A value of **** $P < 0.0001$ seems excessive if the analysis is unpaired. Also related to this panel the authors write: "Compared to EV-control cells, primary CLL cells expressing NOTCH1DPEST consistently showed a higher percentage of cells going through S-phase (Figure 2b)." However, in the way the data is presented a good fraction of CLL samples does not seem to increase the fraction of cells in S-phase upon NOTCH1-ICD transduction. This data must be presented as paired analysis between the same CLL sample comparing EV and NOTCH1-ICD; otherwise, it is not possible to evaluate the effect. The same should be done for the data presented in (D; E; F; and G). Also, the data in panel (A) should consider the paired analysis.

Why were only three patients analyzed in (C)? To which patients does it correspond in (B).

Fig4: In panels (B) to (D) there is the same issue as before if paired analysis is not performed between EV and NOTCH1-ICD it is not possible to interpret the data.

Fig5: (A, C, D, H, I, J) please present paired data.

The authors write "...in keeping with a report showing strong expression of PD-L1 on CLL cells in proliferative centers in lymph nodes. To test whether CLL cells recently egressed from lymph nodes had higher expression levels of PD-L1, we assessed its expression on peripheral blood cells expressing CXCR4dim/CD5bright." Do the CLL primary samples used in (F) display NOTCH1 mutations?

Fig6: (A) The Y axis simply says expression, can the authors elaborate. (F, G) please present paired data.

Fig7: (C) Difficult to appreciate the CD19 stain and its co-stain with Ki67 – can the authors please provide as supp material the individual channels?

(H, I) please present paired data.

The data suggests that NOTCH1 supports proliferation of CD4+ and CD8+ cells. Is this not at odds with the findings in vitro in that NOTCH1-ICD cells inhibit NFAT activation in T cells? Can the authors elaborate.

Also, CD4+ T cells did not show increased PD1 levels, thus on what basis can the authors claim that these cells have a more exhausted phenotype?

Reviewer #2 (Remarks to the Author):

In this article, Mangolini and colleagues studied the effect of NOTCH1 and 2 mutations in CLL and MCL. They used a method that allowed efficient transduction of primary tumor B cells, which enabled them to study the consequences of these mutations, overpassing the need to use less relevant cell lines. From this work, they demonstrated that NOTCH1 mutations induce proliferation of primary malignant B-cells and confirmed the cooperation between NOTCH1 and the BCR signaling. They showed that not only NOTCH mutations drive transcriptional reprogramming but also epigenetic silencing of specific loci. The authors demonstrated that NOTCH mutations are associated with immune escape, through decreased expression of MHCII molecules with a mechanism involving CIITA silencing, and through increased expression of PD-L1 in both CLL and MCL. CIITA is associated with a shorter time to treatment and overall survival in a CLL cohort. Using Hyperion imaging of CLL lymph nodes and PDX model, they unraveled an increased CD4+ T cell infiltration in proliferation centers in association with NOTCH1 mutations.

This work is of great importance, for several reasons. First, the method used to manipulate the primary neoplastic cells is an important breakthrough in the context of B cell malignancies, as until now the available options were very limited, and did not allow the efficient long-term manipulation of primary cells. The use of GaLV or Flv viral envelopes in combination with the co-culture with modified stromal cells, allows an efficient transduction of the primary CLL and MCL cells. Then, the understanding of NOTCH mutations mechanisms of action is crucial to understand their impact on the prognosis of CLL/MCL patients.

Overall, this is an excellent work that deserves publication in Nature Communications. I have few comments and suggestions to further improve the study and the manuscript.

Major comments:

- For all the figures, it would be helpful to have a specific symbol per patient when the samples are matched (e.g. fig2b)

- Figure 2:

b-c: It would be interesting to have CFSE proliferation assay to compare cell-division in EV vs NOTCH1 mut.

g: could you check by WB the activation of the different signaling pathways downstream of the BCR?

- Figure 3:

a: For patients without tri12 or del13q, would you have the same impact of NOTCH1 mutations on the transcriptional program?

It would be better to show the paired samples as in panel f.

I find the cut-off for fold changes low ($\log_2 > 0.5$ in half of samples) compared to usual RNA-sequencing cut-off (> 1)

b: some of the q-value are not significant (> 0.1) in the left plot.

- Figure 4:

g: I don't really see here the impact of Notch activation, it looks more an impact of CIITA (as in f).

Could you also make the KM plots with NOTCH low/high and CIITA low/high separately.

Would need maybe a broader gene signature to define Notch1 low/high (as in 3d?).

Could you please indicate the number of patients per sub-group.

- Figure 5:

i: Did you check also PD-L1 mRNA level? Is there any difference in H3K27ac profile for PD-L1, IFN γ and IFNGR1?

j: Did you find other inflammatory cytokines that could regulate PD-L1 (e.g. IL-6, other IFN, TNF- α) in your RNAseq or proteomic screens? This could be checked by qPCR as in i. Blockade of several receptors might lead to stronger effects.

- Figure 6:

a: It seems that NOTCH2 is more expressed than NOTCH1 in CLL. Is it really the case at the protein level? Why no mutations in CLL? Did you test the effects of NOTCH2dPEST in CLL as in MCL?

- Figure 7:

c: tSNE plots do not bring much information, the different cell clusters are not well separated. More hyperion pictures of the LN with Notch-/+ would be more illustrative (with better resolution). The scale is missing in the magnification.

f: What about Ki67 in CD8+ T cells ?

h: I guess the color code is depending on the donor. Would be good to have the same symbol for the same donor (between EV/N1).

i: please include absolute count for CD4+ and CD8+ T cells

It would be interested to phenotype more precisely these CD4+ T cells in the PDX/hLN. The authors could have included for instance FOXP3 in their panels to discriminate Th/Treg cells.

Minor comments:

- Please explicitate ICD abbreviation at first occurrence.

- Fig4d: miss the statistic

- Fig5i: the lines below the stars are not needed

- Stars/error bars are not consistent between figures

- I would use the term blockade instead of blockage.

Reviewer #3 (Remarks to the Author):

Activating NOTCH mutations in B cell malignancies such as CLL and MCL have been show to correlate with poor clinical outcome. Most NOTCH mutations occur in either NOTCH1 or NOTCH2 and are restricted to the PEST domain or the 3'UTR leading to truncated and more stable versions of NOTCH receptors. Although the putative role of oncogenic NOTCH has been studied using human cell lines and or mouse models it is still not clear how oncogenic NOTCH signaling on a mechanistic or cellular level contributes to a worse disease outcome in these B cell malignancies.

Here the authors tested different viral envelopes for their ability to efficiently infect primary CLL and MCL cells. The Feline Leukemia Virus envelope infected CLL and MCL cells with a relatively high percentage (37 and 39 % respectively, using a GFP expression construct). This methodology was used to express NOTCH1- and NOTCH2-ICD constructs in primary patient derived CLL and MCL cells. The authors provide data showing that NOTCH-IC overexpression facilitates immune escape of malignant B cells by upregulating PD-L1, partially through autocrine interferon- γ secretion. Moreover NOTCH-IC expression causes downregulation of the HLA-class II locus via negative regulation of the transcriptional co-activator CTIIA. These NOTCH-induced immune escape mechanisms were associated with expansion of CD4+ T cells when transduced CLL cells and autologous T cells were

transplanted together into NSG mice.

The findings are in principle interesting and shed potentially new light on the oncogenic function of NOTCH in CLL and MCL.

Following points should be addressed and clarified before the manuscript would be suitable for publication.

1. Figure 1e shows quantification of transduction efficiencies as determined by GFP expression using three different envelope constructs in CLL and MCL primary cells 72 hours post transduction. Left panels should show representative flow cytometric images as stated in the figure legend. However according to the right panel, which shows the quantification and individual data points, the left panel flow cytometric images seem not to be representative as they show extreme data points (e.g. 60% GFP+ cells for GALV envelope, which is the highest data point in the quantification, the mean would be around 25%, similarly 85% GFP+ cells for the FeLV construct, the mean would be 37%). This needs to be corrected. The flow cytometric images should show the average infection rates and not the extremes.

2. Figure 1 e, The EV control is shown as grey bar, but the defining box is shown as white rectangle (minor point).

3. Figure 1g, flow cytometric image of S-Phase with empty vector control seems not be representative as it does not correspond to means as shown in the bar blot on the right panel.

4. One of my major points is the question whether over expression of NOTCH-IC constructs really mimic CLL or MCL cells with NOTCH1 or NOTCH2 PEST domain mutations? These PEST domain mutations result presumably in a moderate increase of ligand-induced NOTCH signaling, while expression of NOTCH-IC constructs will induce a very strong NOTCH signal, that may cause apoptosis in B cells (including CLL and MCL cells). While Figure 1 shows relatively high infectivity of a GFP construct using the FeLV envelope, the infectivity of NOTCH1-iresGFP in primary CLL cells appears to be only 1% (Figure 2c). Overexpression of NOTCH1-IC results in a moderate increase in proliferation as documented by increased cells in S-Phase (Fig. 2b and c). The authors do not comment on apoptosis. While Figure 1 shows that the GFP constructs in combination with the different viral envelopes tested per se does not cause increased apoptosis, overexpression of NOTCH1-IC could. The labeling of the y-axis of Fig 2c is confusing. If the mean % of GFP positive cells is shown then this would mean that only 1% of CLL cells express NOTCH-IC. What is the empty vector control? An empty IRES GFP vector? Why would the infectivity of the control vector be so low? The authors need to clarify this and also comment and show if NOTCH-IC expression causes massive apoptosis in their system. Should that be the case than they may select for CLL and MCL cells that survive the high expression levels of NOTCH-IC, which may explain the low % of CLL cells expressing NOTCH-IC. However, to a certain extent, this also questions the relevance of this model, as for most of the CLL cells NOTCH-IC could just be toxic.

5. What is the endogenous NOCHT expression status of the CLL cells before being transduced with NOTCH1-IC? Primary CLL cells frequently downregulate NOTCH receptor expression, when cultured in vitro.

6. The statistical analysis is only mentioned in materials and methods. It is not clear what statistical analysis is used if only two groups are compared. The STDV seem to overlap quite often for example in Figure 2, nevertheless the minor differences in e.g. MFI of CD49d expression or peak Ca⁺⁺ flux is indicated being significant. The type of statistical test should be mentioned in the figure legend.

7. The authors performed RNA seq and ChIP seq analysis for H3K27ac of NOTCH-IC or vector control expressing CLL cells. They show increased Hes1 expression, which seems to correlate with increased H3K27 acetylation. This is expected. There are no specific comments on other potential genes that might explain the moderate increase in proliferation. Do the authors find increased

expression of genes and or histone marks that might explain increased proliferation? for example Myc-enhancer and or myc expression or E2F target genes?

8. The finding that NOTCH1 expression represses MHC class II genes is interesting. Figure 4c shows cultured CLL cells from 4 donors with endogenous exon 34 mutations of NOTCH1 in the presence or absence of GSI, which resulted in the upregulation of HLA-DR. This is a nice result confirming the outcome of the reciprocal NOTCH1-IC gain of function result shown in Fig. 4b. However, the result of Fig. 4c should be backed up by showing downregulation of NOTCH target genes e.g. Hes1 or DTX1 and that should correlate with upregulation of HLA-DR. This is an important control since GSI can act on many targets.

9. Figures 4 e-g are mentioned wrongly in the main text (page 9 and 10 of the manuscript). The main text refers to these figures as figure 4 i and j and figure 4 e is not mentioned.

10. Figure 5a: Does expression of NOTCH1-IC in primary CLL cells result in downregulation of CD19?, which could explain the reduced luciferase activity of Jurkat cells.

11. Figure 5c: Bar graph of NOTCH1-IC transduced MEC-1 and Hg-3 cells show only 1 data point, suggesting that the experiment has only been performed once?. Is this correct? Moreover, indicating relative MFI is not very informative, what are the real numbers for MFI?

12. Fig. 5d, same comment as point 8.

13. The authors report that NOTCH2-IC also downregulated CT IIA and HLA-class II expression in MCL (page 13). However, the data shown in Figure 6e, f and g show down regulation of CTIIA and HLA-DR as well as upregulation of PD-L1 using NOTCH1-IC. In this context it would make more sense to show data using NOTCH2-IC. It seems not logic to use NOTCH1-IC, if the purpose of the paragraph is to show that NOTCH1 and NOTCH2 seem to induce a similar transcriptional program. Upregulation of PD-L1 by NOTCH2-IC is provided in Suppl. Figure 3. This is ok, but it would be necessary to also show data for NOTCH2-IC mediated downregulating of CTIIA and HLA-DR.

14. The last part of the results section shows that NOTCH1 expressing CLL cells correlate with increased numbers of CD4 T cells. This part is weak and not really conclusive and subject to speculation and overstatements. For example:

To provide further experimental evidence, we injected NOD.Cg-Prkcd scidII2rd tm1Wjl/ Szj (NSG) mice intraperitoneally with isogeneic primary CLL cells, carrying either NOTCH1DPEST or EVcontrol. Prior to this, autologous T cells were isolated with anti-CD3 beads, cultured for 7 days and then co-injected at a ratio of 20:1 (CLL:T cell) (Figure 7g). Assessment of engrafted CLL cells showed a moderate, but not significant, increase in the tumor burden of NOTCH1- transduced cells after 3 weeks (Figure 7h). Importantly, we observed a significant increase of CD4+ T cell in the peritoneal cavity of mice diseased with NOTCH1DPEST-expressing CLL cells, demonstrating that NOTCH1 promotes expansion of CD4+ cells (Figure 7i).

It is not clear how Notch1-IC expressing CLL cells cause an increase in numbers of co-transplanted T cells. Whether this is of physiological importance is not clear either. In my opinion this part could be omitted as it does not add anything to the story.

Mangolini et al; NCOMMS-22-01742-T

General response to all reviewers:

Firstly, we would like to thank all reviewers for their fair and constructive comments. In our experience it is rare to receive such uniformly balanced and constructive reviews. Addressing all their points has clearly helped to improve the manuscript substantially.

The main changes we made to the manuscript are:

1. We decided to change the title; as the reviewers point out, our newly developed method will greatly advance the field and facilitate studying B cell lymphoma. By referring to the method in the title, we hope this will improve visibility and help to distribute this technique to the wider community.
2. We have encrypted each patient with an individual symbol to reflect the matched, pairwise comparison of data-points.
3. We have added a graphical abstract to the figures.

Please find below our detailed response to the reviewers' comments:

Response to reviewer#1.....page 1

Response to reviewer#2.....page 4

Response to reviewer#3.....page 9

We hope with these new data and correction that our manuscript is now acceptable for publication at *Nature communications*.

Point-by-point response to reviewers' comments

Reviewer#1

[...] This reviewer has mainly comments that primarily relate to data presentation.

Fig1: It is advisable that the authors show consistency between the flow plot in (C) with the cumulative data presented. In the cumulative data presented in (C) is each dot representing a different primary sample i.e., biological replicates or are these technical replicates? Was the same primary sample transduced with the various viral envelopes? If yes, it would be important to show the variation for each to evaluate the real difference between the various envelopes.

We apologise that it was not clear that each dot in Figure 1C (right panel) represented an individual biological replicate, using primary tumour cells from different patients. We have now clarified this in the figure legend and encrypted each patient with a unique symbol. This now allows to compare transduction efficacy between different virus envelopes for an individual patient.

In addition, as suggested by this reviewer and reviewer#3, we have replaced dot plots in the same figure (left panel), matching dot blots with the *average* percentage of GFP⁺ cells.

(E) is the data on p21 expression performed on sorted GFP cells? This is not explicit in the figure legend.

We had analysed p21 mRNA expression on sorted GFP⁺ cells. This information is now added to the figure legend.

(G) The data must be presented as paired analysis between the same CLL sample comparing EV and C-MYC otherwise it is not possible to evaluate the effect. The authors may wish to describe in more detail the co-culture system. Once cells are sorted after infection the cells are no longer add to co-culture with feeder cells, correct? E.g., in (C) when it is written in the legend “6 days post transduction” does this correspond to day 13 in the schema in (B)?

We truly regret that it was not clear from our previous manuscript that we indeed compared matched samples (between EV and our gene of interest [NOTCH, MYC, DNTP53]. We have now clarified this in the figure and figure legends throughout all figures and panels by stating: “Each symbol refers to an individual patient sample. Statistical analysis was done on paired samples.”. In addition, we have amended the method section and provide more details on the transduction protocol in the supplementary data. After transduction cells can either be kept on modified stroma cells, transferred to normal stroma cells (not expressing BAFF and IL21 and CD40L) or kept as mono-culture in suspension. We have now added further information to the result section on page 5 and provide data that transfer to un-transduced (non-modified) stroma cells re-instates cell cycle arrest in transduced cells (Supplementary Figure 1a). We believe this information is important for anyone who wants to employ this method.

*Fig2: Could the authors confirm that the statistics presented in (B) are correct? A value of ****P < 0.0001 seems excessive if the analysis is unpaired. Also related to this panel the authors write: “Compared to EV-control cells, primary CLL cells expressing NOTCH1DPEST consistently showed a higher percentage of cells going through S-phase (Figure 2b).” However, in the way the data is presented a good fraction of CLL samples does not seem to increase the fraction of cells in S-phase upon NOTCH1-ICD transduction. This data must be presented as paired analysis between the same CLL sample comparing EV and NOTCH1-ICD; otherwise, it is not possible to evaluate the effect. The same should be done for the data presented in (D; E; F; and G). Also, the data in panel (A) should consider the paired analysis.*

Please see our previous comment. The reviewer is correct that a pairwise comparison is needed, and this is in fact the strength of our newly developed method. We have changed this and most of the other panels such that each patient is encrypted by a unique symbol, allowing the reader to compare individual data points from one patient. We have amended the figure legend accordingly.

Why were only three patients analyzed in (C)? To which patients does it correspond in (B).

Pairwise comparison of NOTCH1^{dPEST} to EV showed a consistent increase in the fraction of cells going through S-phase (n=12, figure 2b). Therefore, we consider N=3 sufficient for the growth curve analyses shown in panel c as it only confirms data depicted in figure 2b. Different patients were used for the experiments depicted in panels b and c. In addition, we have now added CSFE data, further confirming the modest growth advantage of NOTCH-activated cells (Supplementary Figure 2a).

Fig4: In panels (B) to (D) there is the same issue as before if paired analysis is not performed between EV and NOTCH1-ICD it is not possible to interpret the data.

Please see comments above. Paired data are now presented through patient-specific symbols.

Fig5: (A, C, D, H, I, J) please present paired data.

Please see comments above. Paired data are now presented through patient-specific symbols.

The authors write “...in keeping with a report showing strong expression of PD-L1 on CLL cells in proliferative centers in lymph nodes. To test whether CLL cells recently egressed from lymph nodes had higher expression levels of PD-L1, we assessed its expression on peripheral blood cells expressing CXCR4dim/CD5bright.” Do the CLL primary samples used in (F) display NOTCH1 mutations?

The samples used for the analyses in panel F had no known NOTCH1-PEST domain mutations (samples were not routinely screened for 3UTR mutations). We agree that it would be interesting to know

whether the same observation applies to NOTCH1-mutated CLL (as it could be used a biomarker for NOTCH mutations/ activation). Unfortunately, to generate these data, we would require freshly isolated peripheral blood from 3-5 untreated, NOTCH1-mutated patients [as this assay is done from fresh and not frozen material]. Acquiring these samples is likely to take 6-12 months due to the rarity of untreated NOTCH-mutated CLL (despite having a large CLL clinic at our hospital). Therefore, we are unable to provide data on PD-L1 expression on CD5/ CXCR4 sorted cells for NOTCH1-mutated CLL.

Fig6: (A) The Y axis simply says expression, can the authors elaborate. (F, G) please present paired data.

Our apologies for the incomplete labelling of the Y-axis in panel A. The values are microarray normalized values called RMA (robust multi-array average). We have now relabelled the figure. Paired data are now presented for panel f. (Panel 6g does not show paired samples, but a comparison between primary cells and a MCL cell line).

Fig7: (C) Difficult to appreciate the CD19 stain and its co-stain with Ki67 – can the authors please provide as supp material the individual channels?

Following this reviewer's request, we are now providing individual channels for figure 7c, depicted in Supplementary figure 7a.

(H, I) please present paired data.

Paired data are now presented for panels h and i.

The data suggests that NOTCH1 supports proliferation of CD4+ and CD8+ cells. Is this not at odds with the findings in vitro in that NOTCH1-ICD cells inhibit NFAT activation in T cells? Can the authors elaborate.

We agree that the *in vivo* data on NOTCH1-mediated expansion of T cells and our reporter assay seems to be "ad odds". Notably, for the latter experiment, T cell activation was *fully dependent* on the presence of blinatumomab due on the use of allogeneic cells. Therefore, this assay does not reflect a physiological activation of autologous T cells by CLL cells (We employed this assay purely to investigate the role of PD-L1 for T cell activation). Importantly, as shown by others, T cells deficient for NFATc1 and NFATc2 remain their proliferation capacity (Peng et al., Immunity, Vol. 14, 13–20, January, 2001). We therefore do not believe that results from our reporter assay contradict the *in vivo* data obtained from mice and humans (Figure 7).

Also, CD4+ T cells did not show increased PD1 levels, thus on what basis can the authors claim that these cells have a more exhausted phenotype?

Our apologies if the text was unclear in the manuscript. The difference in PD1 expression was shown in CD8⁺ cells only (Figure 7f), hence we stated in the manuscript that CD8⁺ cells, but not CD4 cells, showed a more exhausted phenotype. (We have made this clearer in the result section on page 14 by stating "and indicated that NOTCH1 supports proliferation of CD4⁺ and CD8⁺ cells, with **only** the latter having a more exhausted phenotype")

Reviewer#2:

Overall, this is an excellent work that deserves publication in Nature Communications. I have few comments and suggestions to further improve the study and the manuscript.

Major comments:

- For all the figures, it would be helpful to have a specific symbol per patient when the samples are matched (e.g. fig2b)

Following this reviewer's and others suggestion, we have individualised symbols to allow for a pairwise comparison of data-points throughout the figures.

- Figure 2:

b-c: It would be interesting to have CFSE proliferation assay to compare cell-division in EV vs NOTCH1 mut.

We have now assessed CSFE staining to compare cell divisions in EV vs. NOTCH1^{dPEST}. The results, now also included in the manuscript as Supplementary Figure 2a, show that NOTCH1^{dPEST} transduced cells proliferative slightly faster than EV controls, in keeping with our cell cycle (Figures 2b and 5d) and growth competition data (Figure 2c).

g: could you check by WB the activation of the different signaling pathways downstream of the BCR?

Unfortunately, the amount of protein needed for multiple western blots to assess BCR-kinases exceeds the protein yield obtainable from transduced primary cells. We are therefore unable to provide more data on kinase activation in primary, transduced cells.

- Figure 3:

a: For patients without tri12 or del13q, would you have the same impact of NOTCH1 mutations on the transcriptional program?

To address this question, we have now transduced FISH-normal CLL cells with EV and NOTCH1^{dPEST} and analysed cells for their expression of PD-L1 and HLA-DR. These data indicate that NOTCH1 exerts similar effects on CLL cells without tri12 or del13q (Appendix Figure 1 below). We acknowledge that it still remains possible that differences exist between NOTCH1-regulated genes in these cytogenetically defined subgroups. To fully address this experimentally, larger sequencing studies are needed, which, while being extremely interesting, is outside the scope of the current project and ongoing work.

Appendix Figure 1: Expression of HLA-DR and PD-L1 on cytogenetically defined subgroups of CLL cells, either transduced with N1^{dPEST} or EV. MFI assessed by flow cytometry. N=4 for normal KT FISH patients.

It would be better to show the paired samples as in panel f. I find the cut-off for fold changes low ($\log_2 > 0.5$ in half of samples) compared to usual RNA-sequencing cut-off (> 1) b: some of the q-value are not significant (> 0.1) in the left plot.

In figure 3a we have represented the \log_2 fold change of the gene expression data coming from 13 paired CLL primary cases expressing EV or NOTCH1. As the reviewer 2 requested, in Appendix figure 2 below we are showing the data of the paired samples. In panel A we are showing the NOTCH1^{dPEST}

(N1) and the control (EV) in pairs, with all data normalized together (row z-scores). In Panel B we are showing the same data but the row normalization is done independently in each pair (paired row z-score). CLL cases are highly heterogeneous and, as such, we observe significant basal transcriptional differences between cases. This is exemplified when comparing the normalized EV signal, show in panel A, of all the different cases studied. For example, CLL_01 and CLL_02 have similar levels of expression in the EV but are quite distinct to CLL_06 or CLL_13. Despite this basal heterogeneity, the differential transcriptional changes induced by N1 expression are consistently observed between cases (see panel B). Although it is true that some cases show a more clear N1 induced upregulation (i.e., CLL_01, CLL_02, CLL_03), while others show a more profound downregulation effect (i.e., CLL_13, CLL_09 and CLL_10).

Thus, we believe it is more representative to show the log2FC in the figure, as we initially did. This is representing the differential expression changes between N1 and EV in each case, which takes into account the basal heterogeneity and shows the global effect of NOTCH1^{ΔPest} individually. The reviewer is correct that some of the FDR-q values in panel b are > 0.1. Given the arbitrary cut-off of FDR-q values (and a value of 0.2 may still be significant), we prefer not to change the graph.

A

B

Appendix Figure 2: Heatmap of DE genes following NOTCH1^{ΔPEST} overexpression in CLL cells (n=13). Libraries were generated from mRNA isolated from FACS sorted GFP+ cells 4 days post infection. (A) Normalisation across all samples. (B) Patient-individual normalisation.

- Figure 4:

g: I don't really see here the impact of Notch activation, it looks more an impact of CIITA (as in f). Could you also make the KM plots with NOTCH low/high and CIITA low/high separately. Would need maybe a broader gene signature to define Notch1 low/high (as in 3d?). Could you please indicate the number of patients per sub-group.

Following this reviewer's suggestion, we have reanalysed the data using a broader gene signature for NOTCH target gene. Since this new gene signature did not change the results, we have not amended the figure. Notably, our analyses clearly identified 4 groups, characterised by a high or low NOTCH-gene signature and high or low CIITA levels (see also Supplementary Figure 4). The adverse effect of low expression of CIITA was only observed in NOTCH-high expresser. Therefore, in this cohort CIITA levels alone do not predict for an inferior survival (please see Appendix Figure 3). Further studies are required to assess whether CIITA is an independent prognostic factor for also for overall survival in CLL patients.

Appendix Figure 3: Overall survival of a cohort of patients with a high or low CIITA expression from the CLL-2H trial.

- Figure 5:

i: Did you check also PD-L1 mRNA level? Is there any difference in H3K27ac profile for PD-L1, IFN γ and IFNGR1?

We have now generated the H3K27ac profiles for PD-L1, IFN γ and IFNGR1 as requested. Both PD-L1 and IFNGR1 show a clear H3K27ac peak at the promoter regions in the NOTCH1 ^{Δ Pest} samples compared to controls. These data are now depicted in Supplementary Figure 5d. The IFN γ is not clearly showing this. Interestingly the chromatin surrounding this gene is heterochromatic in CLL and in normal B cells (data from Beekman et al., Nat Med 2018).

With regard to PD-L1 mRNA levels, we observed a small increase (1.4-fold) in mRNA expression induced by NOTCH1-activation (Supplementary Figure 5c), in keeping with the H3K27ac profile for PD-L1 (Supplementary Figure 5d).

j: Did you find other inflammatory cytokines that could regulate PD-L1 (e.g. IL-6, other IFN, TNF- α) in your RNAseq or proteomic screens? This could be checked by qPCR as in i. Blockade of several receptors might lead to stronger effects.

This is a very important point- many thanks. We have analysed our RNAseq data for the expression of IL-6, IL-6R, TNF, INFE, IFNB1, IL-17A, IL-10, IL-27, INGG and IFNA1. The data, depicted in Supplementary Figure 5c, show no significant increase of any of these cytokines by NOTCH1-activation except for IFN γ . We now discuss these findings in our result section on page 11.

- Figure 6:

a: It seems that NOTCH2 is more expressed than NOTCH1 in CLL. Is it really the case at the protein level? Why no mutations in CLL? Did you test the effects of NOTCH2dPEST in CLL as in MCL?

These are all very good questions, which unfortunately are difficult and technically challenging to address. Assessment of surface NOTCH1 expression (which is relevant to understand its biological functions) commonly requires the use of antibodies (for immunoblotting or Flow cytometry). A direct

and quantitative comparison between NOTCH1 and NOTCH2 expression is not possible since these assessments are largely dependent on antibody affinity to their respected epitope, which differs between antibodies even for the same protein. (To address this question, SILAC and mass spectrometry must be employed). Why CLL patients do not acquire NOTCH2 mutations, although they express NOTCH2 mRNA remains unknown. We speculate that mutated NOTCH2 may be detrimental for cells and counter selected. While this is an interesting question, we have not further pursued this experimentally as it is clinically not relevant due to the absence of NOTCH2-mutations in CLL.

- Figure 7:

c: tSNE plots do not bring much information, the different cell clusters are not well separated. More hyperion pictures of the LN with Notch-/+ would be more illustrative (with better resolution). The scale is missing in the magnification.

We have now added a Supplementary Figure 7a, providing single channel images. Contrary to light microscopy images, hyperion images are computationally assembled. Therefore, the resolution of the images is a fixed variable we cannot change for the acquisition of the images. We prefer to leave the tSNE plots in the figure as they illustrate the abundance of the 3 cell populations.

f: What about Ki67 in CD8+ T cells ?

We have observed a trend toward a higher Ki67 expression also in CD8⁺ cells, but this was not significant. We have added this information to the result section. The data are depicted below for the reviewer's benefit (Appendix Figure 4)

Appendix Figure 4: Intensity of cellular signal per given cell was calculated using the HistoCat software. Ki67 signal in CD8 gated cells following tNSE analysis of samples with a positive or negative NOTCH1 nuclear staining

h: I guess the color code is depending on the donor. Would be good to have the same symbol for the same donor (between EV/N1).

Following this reviewer's suggestion, we have now changed the colour codes to symbol codes, in keeping with the remaining data.

i: please include absolute count for CD4+ and CD8+ T cells

We regret that this information is not available. Assessment of absolute counts require the use of counting beads, which were not included in our experiment.

It would be interested to phenotype more precisely these CD4+ T cells in the PDX/hLN. The authors could have included for instance FOXP3 in their panels to discriminate Th/Treg cells.

We agree with this reviewer that it would be interesting to characterise the subset of T cells in the PDX experiments. This is ongoing work and we have yet not completed these studies. For the reviewers' benefit, we here show our first data, indicating that Notch1^{dPEST} in CLL cells skews T cell differentiation to CD4⁺ Th2 cells (CD4⁺ CCR6⁻ CXCR3⁻); Appendix Figure 5.

Notably, and in agreement with reviewer#3, we believe these data are not essential for our current manuscript and we have therefore decided not to include them in the manuscript.

Appendix Figure 5: Quantification of the % of human autologous CD4⁺ T cell subsets in the peritoneal cavity of mice injected with CLL cells with or without NOTCH1^{ΔPEST}

Minor comments:

- Please explicitate ICD abbreviation at first occurrence.
- Fig4d: miss the statistic
- Fig5i: the lines below the stars are not needed
- Stars/error bars are not consistent between figures
- I would use the term blockade instead of blockage.

We are grateful this reviewer spotted these shortcomings. We have incorporated these suggestions in our revised manuscript. Please note that the data depicted in Figure 4d were generated from our RNAseq data and therefore the statistical thresholds were applied as described in the material and method section.

Reviewer#3:

1. Figure 1e shows quantification of transduction efficiencies as determined by GFP expression using three different envelope constructs in CLL and MCL primary cells 72 hours post transduction. Left panels should show representative flow cytometric images as stated in the figure legend. However according to the right panel, which shows the quantification and individual data points, the left panel flow cytometric images seem not to be representative as they show extreme data points (e.g. 60% GFP+ cells for GALV envelope, which is the highest data point in the quantification, the mean would be around 25%, similarly 85% GFP+ cells for the FeLV construct, the mean would be 37%). This needs to be corrected. The flow cytometric images should show the average infection rates and not the extremes.

Following this reviewer's suggestion, we have replaced the flow cytometry images with data more accurately reflecting the mean values depicted on the right panel.

2. Figure 1 e, The EV control is shown as grey bar, but the defining box is shown as white rectangle (minor point).

This has now been corrected.

3. Figure 1g, flow cytometric image of S-Phase with empty vector control seems not be representative as it does not correspond to means as shown in the bar blot on the right panel.

Following this reviewer's suggestion, we have replaced the flow cytometry images with data more accurately reflecting the mean values depicted on the right panel.

4. One of my major points is the question whether over expression of NOTCH-IC constructs really mimic CLL or MCL cells with NOTCH1 or NOTCH2 PEST domain mutations? These PEST domain mutations result presumably in a moderate increase of ligand-induced NOTCH signaling, while expression of NOTCH-IC constructs will induce a very strong NOTCH signal, that may cause apoptosis in B cells (including CLL and MCL cells). While Figure 1 shows relatively high infectivity of a GFP construct using the FeLV envelope, the infectivity of NOTCH1-iresGFP in primary CLL cells appears to be only 1% (Figure 2c).

The transduction efficacy for the NOTCH-IC constructs is in fact the same as for the GFP, between 30-40% on average. Figure 2c shows the relative fold increase of this GFP-positive population (compared to day 3), whereby 1 means no increase of the GFP population and >1 an increase compared to the number of GFP+ cells on day 3. We have now corrected the figure legend to make this clearer as the previous legend was misleading; our apologies for this!

Regarding the signal strength induced by NOTCH-IC compared to endogenous, PEST-domain mutated NOTCH1: We are confident that our NOTCH-constructs do not exert super-physiological activation of NOTCH compared to spontaneously mutated NOTCH1. This is evidenced by (1) similar induction of target genes (Figure 2a), (2) downregulation of PD-L1 expression by GSI in NOTCH1 mutated CLL (Figure 5c), (3) up-regulation of HLA-DR by GSI in NOTCH1 mutated CLL (Figure 4c) and (4) *in vivo* confirmation of PD-L1 expression and (5) T cell expansion (Figure 7).

In addition, while this manuscript was under review, we have generated new data using CRISPR/Cas-mediated deletion of the PEST domain in NOTCH-wild type CLL. These data, depicted here as Appendix Figure 6, demonstrate that deletion of the PEST-domain causes PD-L1 up- and HLA-downregulation, similar to the expression of NOTCH-IC, further confirming that our experimental approach is not overestimating NOTCH-effects. Unfortunately, we are still not able to quantify the efficacy of CRISPR-deletion (which we know from PCR-genotyping is not 100%). Therefore, data in Appendix Figure 6, are the sum-effect of sub-clonal PEST-deleted cells. Further optimisation of this experiment is ongoing and will be reported in the future. The data are depicted for this reviewers' benefit.

Appendix Figure 6: Expression of PD-L1 and HLA-DR on primary CLL cells 3 days after nucleofection of CAS12a and control guide RNAs or gRNAs targeting the PEST domain of endogenous *NOTCH1*.

Overexpression of NOTCH1-IC results in a moderate increase in proliferation as documented by increased cells in S-Phase (Fig. 2b and c). The authors do not comment on apoptosis. While Figure 1 shows that the GFP constructs in combination with the different viral envelopes tested per se does not cause increased apoptosis, overexpression of NOTCH1-IC could. The labeling of the y-axis of Fig 2c is confusing. If the mean % of GFP positive cells is shown then this would mean that only 1% of CLL cells express NOTCH-IC. What is the empty vector control? An empty IRES GFP vector? Why would the infectivity of the control vector be so low? The authors need to clarify this and also comment and show if NOTCH-IC expression causes massive apoptosis in their system. Should that be the case than they may select for CLL and MCL cells that survive the high expression levels of NOTCH-IC, which may explain the low % of CLL cells expressing NOTCH-IC. However, to a certain extent, this also questions the relevance of this model, as for most of the CLL cells NOTCH-IC could just be toxic.

We agree with the reviewer that NOTCH-induced apoptosis is an important point we had not discussed in our previous submission. We now analysed this and our data indicate only a very moderate induction of apoptosis by NOTCH1-IC. This effect is outcompeted by the proliferative advantage of NOTCH-activation as indicated by an increase in the number of GFP+ cells in our growth competition assay (Figure 2c). We have incorporated this data in the main manuscript (Figure 2d) and discuss this accordingly. Many thanks for raising this important point that we missed.

5. What is the endogenous NOTCH expression status of the CLL cells before being transduced with NOTCH1-IC? Primary CLL cells frequently downregulate NOTCH receptor expression, when cultured *in vitro*.

Following this reviewers' comment, we have analysed surface NOTCH1 expression in untransduced primary CLL cells cultured *in vitro* over a period of 48 hours. The data, depicted in Appendix Figure 7, show a constitutive expression of NOTCH1 (in keeping with published data; PMID 22829975), which did not significantly change over time. We apologise that we have not been able to identify the paper the reviewer refers to, demonstrating downregulation of NOTCH1 *in vitro* cultures and therefore cannot comment on the reason for this apparent discrepancy.

Appendix Figure 7: NOTCH1 cell surface expression over 48 hours *in vitro* culture. NOTCH1 expression was assessed by Flow Cytometry by co-staining for NOTCH1, CD19 and CD5 (n=3).

6. The statistical analysis is only mentioned in materials and methods. It is not clear what statistical analysis is used if only two groups are compared. The STDV seem to overlap quite often for example in Figure 2, nevertheless the minor differences in e.g. MFI of CD49d expression or peak Ca⁺⁺ flux is indicated being significant. The type of statistical test should be mentioned in the figure legend.

We apologise that the information was not provided in more detail. We have used a one-way ANOVA test followed by paired t-tests. Data are depicted as mean \pm SEM. Matched samples were analysed, which is indeed a strength of our newly developed transduction system. The information is provided in the material and method section.

7. The authors performed RNA seq and ChIP seq analysis for H3K27ac of NOTCH-IC or vector control expressing CLL cells. They show increased Hes1 expression, which seems to correlate with increased H3K27 acetylation. This is expected. There are no specific comments on other potential genes that might explain the moderate increase in proliferation. Do the authors find increased expression of genes and/or histone marks that might explain increased proliferation? for example Myc-enhancer and/or myc expression or E2F target genes?

This is an interesting point; to address it, we have now analysed RNA expression of cell cycle genes and canonical Myc-targets. These data are included in the main manuscript (Supplementary Figure 3c). We did not find a NOTCH-dependent increased expression of any of those genes, suggesting that the proliferative advantage of NOTCH-activated cells is not based on transcriptional regulation of these genes, but rather mediated by post-transcriptional mechanism.

8. The finding that NOTCH1 expression represses MHC class II genes is interesting. Figure 4c shows cultured CLL cells from 4 donors with endogenous exon 34 mutations of NOTCH1 in the presence or absence of GSI, which resulted in the upregulation of HLA-DR. This is a nice result confirming the outcome of the reciprocal NOTCH1-IC gain of function result shown in Fig. 4b. However, the result of Fig. 4c should be backed up by showing downregulation of NOTCH target genes e.g. Hes1 or DTX1 and that should correlate with upregulation of HLA-DR. This is an important control since GSI can act on many targets.

We agree that this is an important control for the experiment depicted in figure 4c. Analyses of NOTCH-target genes indeed showed their downregulation by GSI-treatment. The data is now included in the manuscript and can be found in Supplementary Figure 4b.

9. Figures 4 e-g are mentioned wrongly in the main text (page 9 and 10 of the manuscript). The main text refers to these figures as figure 4 i and j and figure 4 e is not mentioned.

This has now been corrected—many thanks for pointing this out.

10. Figure 5a: Does expression of NOTCH1-IC in primary CLL cells result in downregulation of CD19?, which could explain the reduced luciferase activity of Jurkat cells.

The reviewer raises an excellent question. Following their concerns, we have analysed CD19 expression in NOTCH1-ICD transduced cells. Compared to the EV control, NOTCH1 indeed causes a significant down-regulation of surface CD19. These data have now been included in the main manuscript (Figure 5i) and referenced in the abstract as we believe this information is important from a clinical perspective as it indicates that patients with NOTCH-mutations may be less suitable for bispecific antibodies or CAR-T cell therapies (utilising this antigen).

To address to what extent this down-regulation of CD19 contributes to the reduced luciferase activity of Jurkat cells, we have now repeated this experiment in the presence of the PD-L1 inhibitor durvalumab. Blockade of PD-L1 rescued the NOTCH1-mediated reduction of Luciferase activity, indicating that CD19-expression was not a limiting factor in this experiment (likely Blinatumumab was excessively available in this *in vitro* experiment).

11. Figure 5c: Bar graph of NOTCH1-IC transduced MEC-1 and Hg-3 cells show only 1 data point, suggesting that the experiment has only been performed once?. Is this correct? Moreover, indicating relative MFI is not very informative, what are the real numbers for MFI?

We regret that it was not clear from our presentation and experiments with MEC-1 and Hg-1 were done on 3 independent replicates. We now clarify this in the figure legends and figure. Furthermore, we have chosen to show the relative MFI since the baseline expression of PD-L1 varies between patients, in keeping with other reports (PMID 33931470). Notably, we observed a reduced Luciferase activity in all experiments with T reporter cells cultured with NOTCH1^{dPEST} CLL cells (compared to EV) (Figure 5j), indicating that the NOTCH-dependent up-regulation of PD-L1 has biological effects despite variations in the baseline expression of PD-L1.

12. Fig. 5d, same comment as point 8.

We agree that this is an important control for the experiment depicted in figure 5d. Analyses of NOTCH-target genes indeed showed a downregulation by GSI-treatment (Supplementary Figure 4b).

13. The authors report that NOTCH2-IC also downregulated CTIIA and HLA-class II expression in MCL (page 13). However, the data shown in Figure 6e, f and g show down regulation of CTIIA and HLA-DR as well as upregulation of PD-L1 using NOTCH1-IC. In this context it would make more sense to show data using NOTCH2-IC. It seems not logic to use NOTCH1-IC, if the purpose of the paragraph is to show that NOTCH1 and NOTCH2 seem to induce a similar transcriptional program. Upregulation of PD-L1 by NOTCH2-IC is provided in Suppl. Figure 3. This is ok, but it would be necessary to also show data for NOTCH2-IC mediated downregulating of CTIIA and HLA-DR.

We regret that NOTCH2-mediated downregulation of CTIIA and HLA-DR was not obvious in our previously submitted manuscript. Downregulation of these genes by NOTCH2 is depicted in figure 6d, showing similar effects than observed for NOTCH1. We confirmed reduced expression of HLA-DR by NOTCH2 in primary MCL-cells by FACS (Supplementary Figure 6a).

14. The last part of the results section shows that NOTCH1 expressing CLL cells correlate with increased numbers of CD4 T cells. This part is weak and not really conclusive and subject to speculation and overstatements. For example: To provide further experimental evidence, we injected NOD.Cg-Prkcd scid1l2rd tm1Wjl/ Szj (NSG) mice intraperitoneally with isogeneic primary CLL cells, carrying either NOTCH1DPEST or EV control. Prior to this, autologous T cells were isolated with anti-CD3 beads, cultured for 7 days and then co-injected at a ratio of 20:1 (CLL:T cell) (Figure 7g). Assessment of engrafted CLL cells showed a moderate, but not significant, increase in the tumor burden of NOTCH1-transduced cells after 3 weeks (Figure 7h). Importantly, we observed a significant increase of CD4+ T cell in the peritoneal cavity of mice diseased with NOTCH1DPEST-expressing CLL cells, demonstrating that NOTCH1 promotes expansion of CD4+ cells (Figure 7i). It is not clear how Notch1-IC expressing CLL cells cause an increase in numbers of co-transplanted T cells. Whether this is of physiological importance is not clear either. In my opinion this part could be omitted as it does not add anything to the story.

We agree with this reviewer that the mechanisms of NOTCH1 mediated skewing of T cells remains unknown and has not been experimentally addressed in our current project. Whether this T cell skewing contributes to disease progression is also not entirely clear, but we believe it is reasonable to speculate that the net effect of this T cell skewing is pro-tumorigenic, in keeping with published data [PMID: 23933259 and PMID:21385850]. Despite the absence of mechanistic data, we prefer to keep the data in the manuscript, as they further validate our experiment due to the correlative data from patients (Figure 7f). In addition, we hope that the data will stimulate other groups to ignite further work on NOTCH1-mediate skewing of T cells.

REVIEWERS' COMMENTS

Reviewer #1 (Remarks to the Author):

The authors have addressed most of the concerns of this reviewer.

Reviewer #2 (Remarks to the Author):

The authors revised their manuscript according to the recommendations of the 3 reviewers. They performed adequate new experiments to answer the questions or provide sound justifications. In my opinion, the article is now suitable for publication in Nature Communications and will undoubtedly generate the interest of the community.

Reviewer #3 (Remarks to the Author):

The authors have replied satisfactorily to all my comments and questions. They have done a nice job. The manuscript improved significantly, and in my opinion is interesting and now suitable for publication.